# AAV gene therapy rescues hearing and balance in a model of *CLIC5* deafness

Roni Hahn [1], Shahar Taiber [1], Olga Shubina-Oleinik [2], Gwenaëlle S G Géléoc [2], Jeffrey R Holt [2] & Karen B Avraham [1✉]

## Abstract

Adeno-associated virus-based gene therapy offers a promising treatment paradigm for inner ear diseases; however, the genetic heterogeneity of hereditary deafness requires gene-specific strategies and optimization of current approaches to identify the range of treatable conditions and improve therapeutic outcomes. To consider the therapeutic potential for a hearing loss gene not previously explored, we investigated the gene encoding the chloride intracellular channel protein CLIC5, mutations in which lead to DFNB103 in humans and deafness and circling behavior in a *Clic5*-deficient mouse model. In this study, we utilized two constructs to deliver the wild-type *Clic5* coding sequence into *Clic5*-deficient mice: single-stranded and self-complementary adeno-associated virus, the latter known for rapid onset of transgene expression. We report a robust restoration of CLIC5 expression using either construct, including prevention of morphological degeneration and preserving auditory and vestibular function. Interestingly, the self-complementary construct achieved comparable functional recovery to single-stranded construct but at a lower titer. These findings highlight the potential of self-complementary adeno-associated virus to reduce dose requirements, minimize toxicity and broaden clinical utility for inner ear therapies.

**Keywords** AAV; Deafness; Gene Therapy; Stereocilia; Vestibular Dysfunction
**Subject Category** Genetics, Gene Therapy & Genetic Disease

## Introduction

Hearing loss, the most common sensory disorder worldwide, has long been explored as a condition amenable to gene therapy (Duhon et al, 2024; Hahn and Avraham, 2023). Advances in understanding disease-related gene functions, combined with improved technologies such as gene replacement using adeno-associated virus (AAV) vectors, have finally made gene therapy a clinical reality for deafness (Akil et al, 2019; Al-Moyed et al, 2019; Brigande, 2024; Lv et al, 2024; Qi et al, 2024a; Smith et al, 2024).

However, the genetic heterogeneity of hearing loss remains a significant challenge, requiring gene-specific analysis and optimization of current approaches for broader treatment applications.

Variants in the *CLIC5* gene, encoding Chloride Intracellular Channel 5, have been found to cause autosomal recessive hearing loss in humans, along with vestibular dysfunction, observed in some individuals and a related mouse model (Gagnon et al, 2006; Seco et al, 2015; Wonkam-Tingang et al, 2020). The CLIC5 protein is expressed in hair cells (HCs) and localized at the base of the stereocilia (Li et al, 2021; Wonkam-Tingang et al, 2020). Together with other hair-cell-expressing proteins such as RDX, PTPRQ, and TPRN, CLIC5 likely plays a role in stabilizing membrane-cytoskeletal attachments at the base of the hair bundle (Qi et al, 2024b; Salles et al, 2014). CLIC5 deficiency leads to defects in the structure and organization of stereocilia, including disarrayed and fused bundles (Salles et al, 2014). Beyond its role in the inner ear, CLIC5 is essential for ciliary functions and is involved in myogenesis, muscle regeneration, and membrane fusion events (Manori et al, 2024; Ott et al, 2023; Suzuki et al, 2018).

Efforts to enhance therapies for inner ear disorders have involved strategies including optimizing capsids such as the synthetic capsid AAV9-PHP.B, using specialized promoters such as Myo15, and improving delivery methods through the utricle route (Deverman et al, 2016; Gyorgy et al, 2019; Hu et al, 2024; Lee et al, 2020; Taiber et al, 2021; Wang et al, 2024). Self-complementary (sc) AAV, recognized for fast onset of transgene expression, offers the potential to further improve therapeutic outcomes (McCarty, 2008; McCarty et al, 2001). Its structure, consisting of two complementary strands separated by a modified ITR, allows scAAV to fold upon itself and bypass the requirement for second strand synthesis by host cells (McCarty, 2008; McCarty et al, 2003; Petersen-Jones et al, 2009). Since this is a rate-limiting step, scAAV enables a faster and more robust onset of transgene expression relative to traditional single-stranded (ss) AAV vectors (Ferrari et al, 1996; Fisher et al, 1996; Hacker et al, 2005; McCarty et al, 2003; Petersen-Jones et al, 2009; Ren et al, 2005; Wang et al, 2005). This outcome has also been observed in the inner ear with AAV vectors driving green fluorescent protein (GFP) expression in cochlear explants and in vivo with cochlear injections (Casey et al, 2020; Maguire and Corey, 2020).

In this study, we aimed to replace the mutant *Clic5* gene in a mouse model (c.680 T > C) of DFNB103 recessive deafness and vestibular

[1]Department of Human Molecular Genetics & Biochemistry, Gray Faculty of Medical & Health Sciences and Sagol School of Neuroscience, Tel Aviv University, Tel Aviv 6997801, Israel. [2]Department of Otolaryngology & Neurology, Boston Children's Hospital, Harvard Medical School, Boston, MA 02115, USA. ✉E-mail: karena@tauex.tau.ac.il

dysfunction, using ssAAV or scAAV vectors to drive wild-type (WT) *Clic5* transgene expression. Our results demonstrate the feasibility of restoring CLIC5 expression, reducing cell death and morphological degeneration, and rescuing auditory and vestibular function in *Clic5*-deficient mice. Furthermore, we found that a lower dose of scAAV yields an equivalent therapeutic outcome relative to ssAAV, suggesting that scAAV could be a more efficient and effective gene therapy vector for hearing and balance disorders.

# Results

## Characterization of *Clic5*-deficient mice

The *Clic5* mutant mouse model we used for this study has an ENU-induced single nucleotide substitution c.680 T > C in the *Clic5* gene, leading to a change of a highly conserved phenylalanine to serine p.(Phe318Ser) (Jax #005329). To characterize this mutant model, hereafter referred to as *Clic5*$^{-/-}$, we investigated its auditory and vestibular phenotypes.

Auditory function was assessed by recording auditory brainstem response (ABR) and distortion product otoacoustic emissions (DPOAE) in WT, heterozygous, and homozygous *Clic5* mice at 4 and 8 weeks. ABR thresholds for the heterozygous mice were found to be similar to those of the WT controls, with no significant differences observed across all frequencies up to 8 weeks, except for 35 kHz (Fig. 1A,B; $P \geq 0.14$ for all frequencies at 4 and 8 weeks). In contrast, *Clic5*$^{-/-}$ mice exhibited ABR thresholds shifted by ~40 dB at 4 weeks, and no response was detected at 8 weeks (Fig. 1A,B). DPOAE results also indicated significant differences of 20 to 50 dB between the *Clic5*$^{-/-}$ mice and both *Clic5*$^{+/+}$ and *Clic5*$^{+/-}$ groups, except for 6 kHz (Fig. 1C; $P \geq 0.13$). There were no differences between *Clic5*$^{+/+}$ and *Clic5*$^{+/-}$ mice, suggesting that *Clic5*$^{+/-}$ mice may serve as a suitable control group for subsequent experiments.

Vestibular function was evaluated with open-field and rotarod assays conducted concurrently with auditory assessments. Abnormal balance and behavior were observed in *Clic5*$^{-/-}$ mice, as indicated by circling behavior and hyperactivity at 4 weeks (Fig. 1D,E). Poor performance on the rotarod assay was noted for the homozygous mice, with a significant difference from WT and heterozygous groups only at 8 weeks (Fig. 1F,G). These results suggest a potentially progressive nature of vestibular dysfunction in *Clic5*$^{-/-}$ mice.

Morphological changes were studied using scanning electron microscopy (SEM). At postnatal day 17 (P17), hair cell abnormalities were evident in the auditory system, with outer hair cell (OHC) bundles appearing fused and disorganized, particularly in the apical region (Fig. 1H,I). Inner hair cells (IHCs) exhibited extensive hair bundle fusion across all cochlear regions. No morphological changes were observed at P14, suggesting that the onset of hair cell defects occurs between P14 and P17 (Fig. 1I). In the vestibular system, vestibular hair cells (VHCs) of the utricle displayed elongated hair bundles as early as P15, and by P17, thickened hair bundles were also observed (Fig. 1J).

## Gene therapy restores CLIC5 expression in the auditory system

To restore CLIC5 expression in the *Clic5*$^{-/-}$ mouse mutant, we first created a ssAAV2/9-PHP.B vector encoding the WT form of *Clic5*

(ssAAV.*Clic5*). The expression was driven by the chicken beta-actin-derived promoter (CB6), known for its stable expression in different cell types, a turboGFP fluorescent reporter, a 3XFLAG tag, and a woodchuck hepatitis virus post-transcriptional regulatory element (WPRE) (Marcovich et al, 2022; Rashnonejad et al, 2016). A total of 1.2 μl of ssAAV.*Clic5* (titer $1.69 \times 10^{14}$ gc/ml) was injected into the inner ear of *Clic5*$^{-/-}$ mice via the utricle route at P0, and inner ears were harvested at 12 weeks (Lee et al, 2020). *Clic5*$^{-/-}$ mice injected with ssAAV.*Clic5* demonstrated robust expression of CLIC5 protein in transduced IHCs and OHCs, while no expression was detected in *Clic5*$^{-/-}$ untreated control (Fig. 2A). The expression of CLIC5 in the treated hair cells was concentrated at the base of the hair bundle, consistent with the pattern seen in *Clic5*$^{+/-}$ mice expressing endogenous functional CLIC5 (Fig. 2B,C). In addition, CLIC5 staining co-localized with the FLAG tag staining, indicating specificity of the anti-CLIC5 antibody used for this study (Fig. 2C).

RDX and TPRN, similar to CLIC5, are proteins expressed in HCs and are normally localized at the base of the stereocilia. Previous studies have shown that the absence of CLIC5 disrupts this localization, resulting in the distribution of RDX and TPRN along the stereocilia, which may contribute to the progressive degeneration of the hair bundle structure in *Clic5*$^{-/-}$ mice (Salles et al, 2014) (Fig. 2D,E). To test whether restoring CLIC5 expression could restore the proper localization of these proteins, *Clic5*$^{-/-}$ mice were injected with ssAAV.*Clic5* (titer $1.69 \times 10^{13}$ gc/ml) at P0 and harvested 8 weeks post injection. RDX expression was concentrated at the base of the hair bundle in treated mice, similar to the pattern observed in *Clic5*$^{+/+}$ control (Fig. 2D). In contrast, TPRN was mislocalized along the entire length of the stereocilium, rather than being concentrated at the base (Fig. 2E). A similar mislocalization pattern was reported by Qi et al, where AAV-mediated TPRN overexpression disrupted its typical localization (Qi et al, 2024b).

## Gene therapy restores the morphology of hair cells in the auditory system

To assess the impact of *Clic5* gene replacement on hair cell morphology and survival, inner ears of mice injected with ssAAV.*Clic5* were harvested at 12 weeks and analyzed using a fluorescent actin label and myosin VIIa antibody. In the untreated *Clic5*$^{-/-}$ mice, staining of hair bundle actin with phalloidin revealed severely disorganized and significantly elongated hair bundles compared to control heterozygous mice, especially in IHCs (Fig. 3A,B; $P < 0.0001$). In contrast, hair bundles of the treated group appeared similar to those of the *Clic5*$^{+/-}$ control mice, organized into distinct rows and displaying uniform and consistent length (Fig. 3A,B). Quantification of HC survival indicated that CLIC5 deficiency primarily caused OHC loss, which was most severe in the basal cochlear region (32 kHz) (Fig. 3C). The treatment with ssAAV.*Clic5* significantly improved OHC survival, resulting in more OHCs observed across all three regions, compared to untreated *Clic5*$^{-/-}$ mice ($P \leq 0.0001$). For IHCs, the absence of CLIC5 had minimal impact on their survival, however significant differences in IHC number for the three groups were observed in the 32 kHz region (Fig. 3D; $P \leq 0.0055$).

For a more detailed examination of stereocilia architecture, SEM was used. Consistent with the confocal microscopy findings, we observed hair cell loss across various cochlear regions in the mutant mice (Fig. 3E). In these mutants, the stereocilia staircase pattern

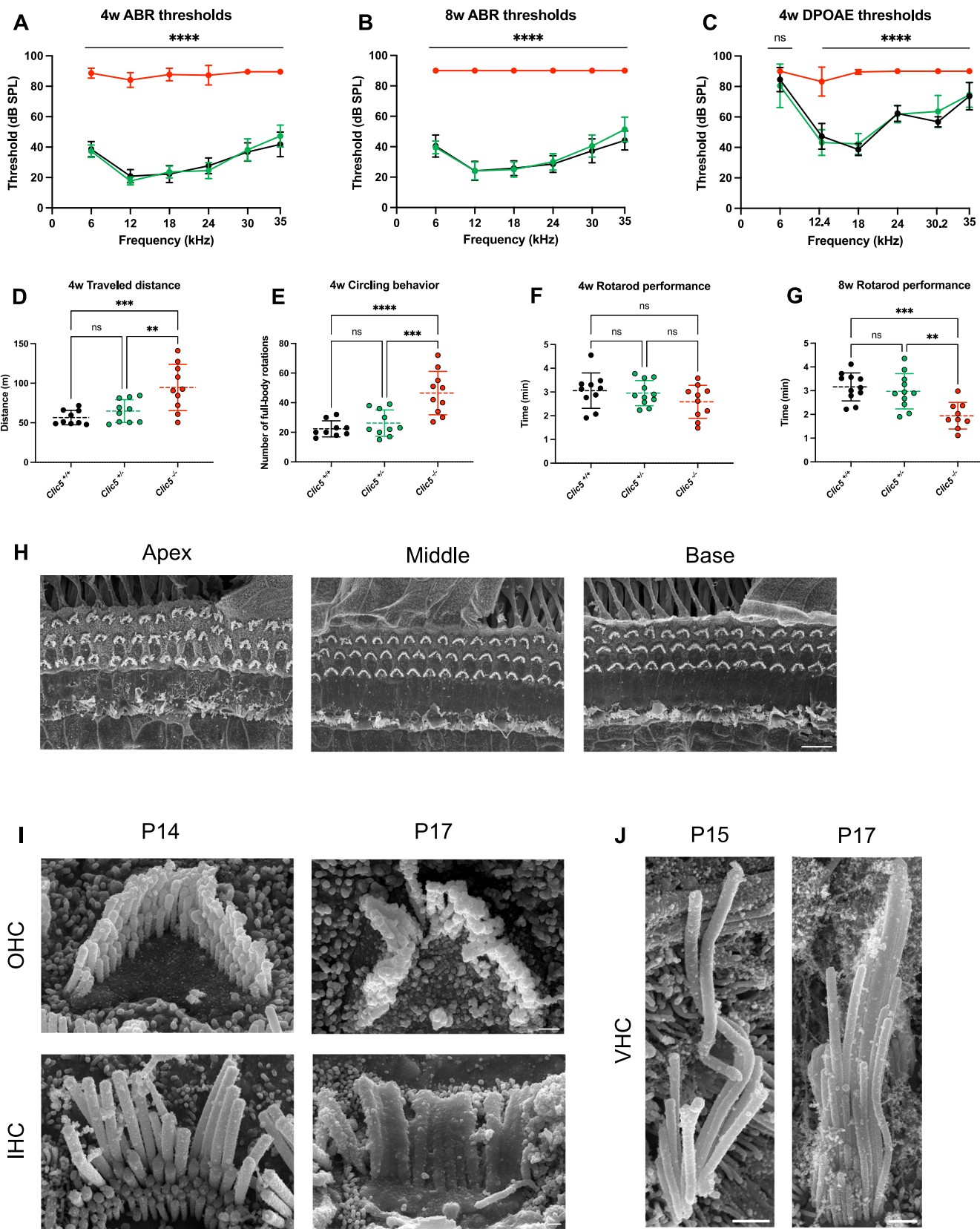

**Figure 1. Functional and morphological characterization of *Clic5* mice.**

(A) ABR thresholds at 4 weeks of WT ($n = 11$), *Clic5*$^{+/-}$ ($n = 11$), and *Clic5*$^{-/-}$ mice ($n = 11$). (B) ABR thresholds at 8 weeks of WT ($n = 11$), *Clic5*$^{+/-}$ ($n = 11$), and *Clic5*$^{-/-}$ mice ($n = 11$). (C) DPOAE thresholds at 4 weeks of WT ($n = 11$), *Clic5*$^{+/-}$ ($n = 11$), and *Clic5*$^{-/-}$ mice ($n = 11$). (D) Distance traveled in the open-field test performed at 4 weeks of WT ($n = 9$), *Clic5*$^{+/-}$ ($n = 10$), and *Clic5*$^{-/-}$ mice ($n = 10$). (E) Quantification of circling behavior testing at 4 weeks of WT ($n = 9$), *Clic5*$^{+/-}$ ($n = 10$), and *Clic5*$^{-/-}$ mice ($n = 10$). (F) Each animal's average time remained balanced on the rotarod apparatus at 4 weeks, WT ($n = 10$), *Clic5*$^{+/-}$ ($n = 11$), and *Clic5*$^{-/-}$ mice ($n = 10$). (G) Each animal's average time remained balanced on the rotarod apparatus at 8 weeks, WT ($n = 11$), *Clic5*$^{+/-}$ ($n = 11$), and *Clic5*$^{-/-}$ mice ($n = 9$). (H) Representative scanning electron microscopy images of IHCs and OHCs at P17 at different regions of the cochlea of *Clic5*$^{-/-}$. (I) High-magnification images of outer and inner hair bundles from the cochlear apex of *Clic5*$^{-/-}$ at P14 and P17. (J) High-magnification images of VHCs from the utricle of *Clic5*$^{-/-}$ at P15 and P17. Data information: Statistical tests were two-way ANOVA with Holm–Sidak correction for multiple comparisons for (A–C) and one-way ANOVA followed by Tukey correction for multiple comparisons for (D–G). Plots show mean ± SD. ns = not significant, **$P < 0.01$, ***$P < 0.001$, ****$P < 0.0001$ (Exact *P* values are provided in Appendix Table S1). Scale bars = 10 μm for (H), 0.5 μm for (I), and 1 μm for (J). Source data are available online for this figure.

was disrupted, and extensive stereocilia fusion was observed in both outer and inner HCs (Fig. 3F). However, in *Clic5*$^{-/-}$ mice treated with ssAAV.*Clic5*, OHC bundles were well organized in three rows with a normal staircase pattern, and IHCs maintained a consistent length similar to the control group (Fig. 3E,F).

## Gene therapy rescues hearing in mutant mice

To evaluate the therapeutic effect of gene replacement therapy on auditory function, ABRs and DPOAEs were recorded at different time points in mice that received injections of ssAAV.*Clic5* at P0. ABR results of untreated mutant mice exhibited almost complete hearing loss by 4 weeks of age and no response by 8 weeks, with no detectable ABR waveforms up to 90 dB (Fig. 4A–C). Mice injected with ssAAV.*Clic5* showed rescued hearing thresholds at all frequencies at 4 weeks (Fig. 4B). At 8 weeks, low and mid-range thresholds remained reduced, while no significant rescue was observed at 30 and 35 kHz (Fig. 4C; $P \geq 0.1355$). At 12 weeks, treated mice exhibited ABR thresholds significantly lower for the 6–24 kHz range compared to the untreated group (*Clic5*$^{-/-}$ mice) but were significantly higher than those of the control group (*Clic5*$^{+/-}$ mice) (Fig. 4D). DPOAE assessments at 4 weeks revealed significant recovery of thresholds at the low frequencies of 12.4 and 18 kHz, indicating rescue of OHC function with ssAAV vector (Fig. 4E; $P < 0.0001$). Over time, the hearing sensitivity of the mutant mice treated with ssAAV.*Clic5* mainly remained stable at low frequencies (Fig. 4F). To validate that the observed rescue is achieved only with an AAV encoding CLIC5, we used a control AAV encoding GFP, which did not show a similar rescue (Fig. EV1A–C). In addition, no effect on hearing function was observed in injected *Clic5*$^{+/-}$ mice with ssAAV.*Clic5* (Fig. EV2A–D).

Another approach to evaluate auditory functional rescue and its durability in mice is the fear conditioning assay. Twelve-week-old mice were exposed to a 6 kHz sound paired with a mild shock to condition them to associate the sound with an aversive experience. The next day, the tone was played without the shock, and the freezing response was measured to indicate sound perception. While *Clic5*$^{-/-}$ mice exhibited similar freezing rates regardless of the presence of the stimulus, mice injected with ssAAV.*Clic5* showed a significant increase in freezing behavior when the tone was presented compared to before it appeared, similar to the control *Clic5*$^{+/-}$ mice (Fig. 4G; $P = 0.0265$). These results indicate that ssAAV not only improved ABR and DPOAE thresholds but also rescued sound perception and facilitated associative learning in the treated mice.

## Gene therapy restores CLIC5 expression and morphology of vestibular hair cells

A key characteristic of the *Clic5* mutant mouse phenotype is vestibular dysfunction, which translates to repetitive circling behavior, abnormal head movements, and overall balance dysfunction. These phenotypes appear to result from the absence of CLIC5 expression in VHCs. The utricle injection method has been previously proven to be an effective method for transducing all six sensory organs within the inner ear of neonatal mice (Lee et al, 2020). Therefore, we examined whether expression of CLIC5 could also be restored in utricular hair cells. In the control *Clic5*$^{+/-}$ group, VHCs showed staining of CLIC5 protein at the base of the stereocilia, similar to its localization pattern in cochlear hair cells (Fig. 5A). In contrast, no expression of CLIC5 was seen in the utricle of *Clic5*$^{-/-}$ mice. When ssAAV.*Clic5* vectors were injected into the inner ears of *Clic5*$^{-/-}$ mice, CLIC5 expression was successfully restored and concentrated at the base of the stereocilia in utricular hair cells (Fig. 5A). Similar to cochlear hair cells of *Clic5*$^{-/-}$ mice, VHCs also showed progressive degeneration, characterized by elongated stereocilia bundles at 12 weeks of age (Fig. 5A,B). Following the injection of ssAAV.*Clic5*, vestibular hair bundles appeared uniformly sized, similar to those observed in control *Clic5*$^{+/-}$ mice (Fig. 5B). Additionally, the number of VHCs, which were reduced in mutant mice, was restored to near control levels following treatment with ssAAV vector (Fig. 5C). Visualized using SEM, VHCs from the utricle of treated mice showed organized hair bundles with rows of increasing height, similar to those from cells of control heterozygous group (Fig. 5D). In contrast, the morphology of utricular hair cells in untreated *Clic5*$^{-/-}$ displayed fused giant hair bundles and lacked a regular staircase pattern (Fig. 5D,E).

## Gene therapy rescues the vestibular function of the mutant mice

To evaluate the therapeutic effect of *Clic5* gene replacement on vestibular function, *Clic5*-deficient mice were tested for different behavioral assays. The balance of the different groups was evaluated using an accelerating rotarod, with the latency to fall being recorded over five trials. Untreated *Clic5*$^{-/-}$ mice performed poorly on the rotarod, whereas those injected with ssAAV.*Clic5* were able to remain on the rotating rod for a longer duration, leading to a significant difference between the groups at 8 weeks of age (Fig. 5F,G; $P = 0.0181$). In addition, no balance defects were observed in *Clic5*$^{+/-}$ mice injected with ssAAV.*Clic5*, excluding

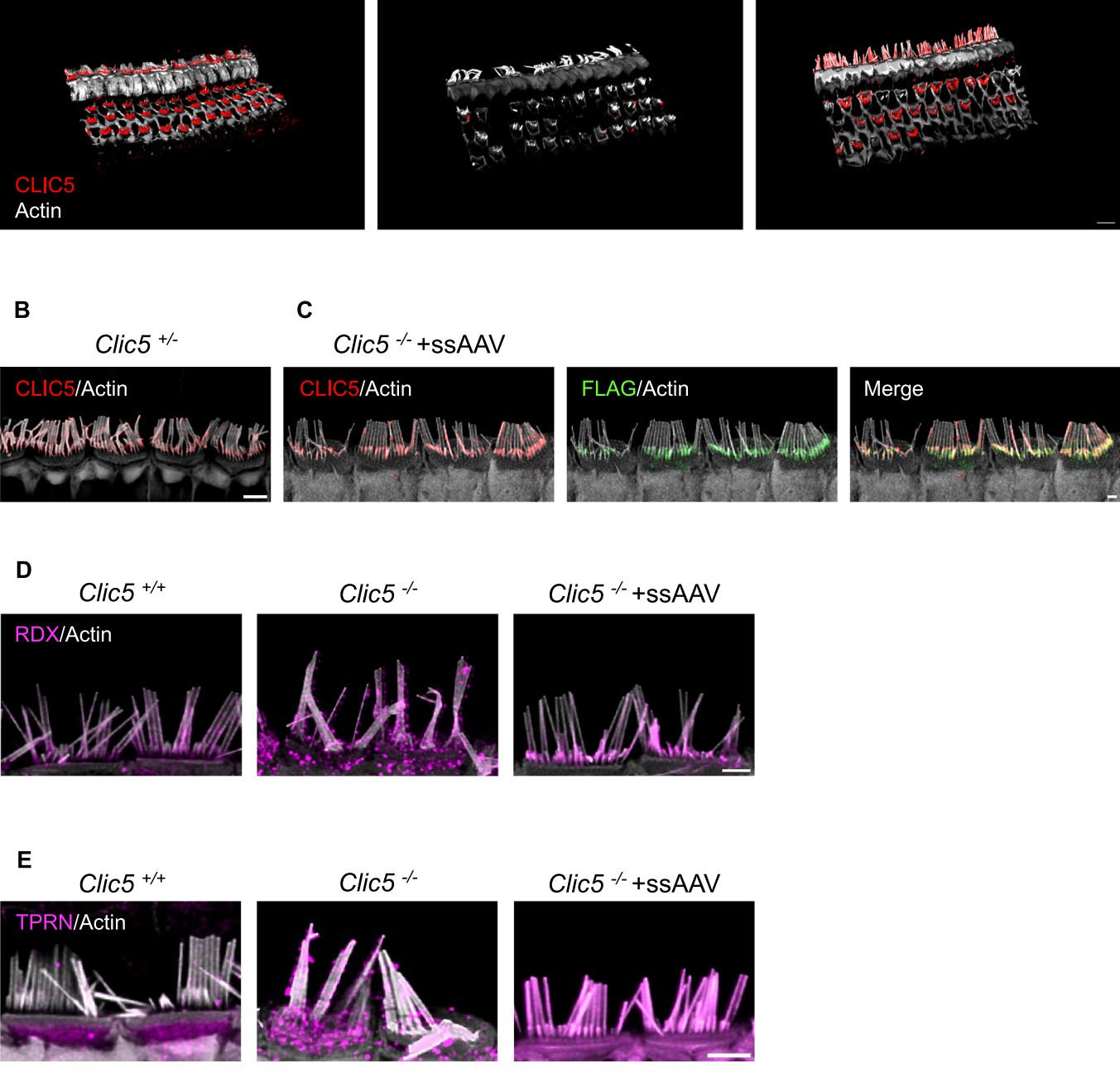

Figure 2. Single-stranded AAV restores CLIC5 expression in the auditory system.

(A) 3D surface projection of IHCs and OHCs from the cochlear 8 kHz region of Clic5+/−, Clic5−/−, and Clic5−/− injected with ssAAV.Clic5 at 12 weeks, stained with a CLIC5 (red) and actin (gray). (B) Representative confocal images of IHCs from the apical turn of Clic5+/− stained with a CLIC5 (red) and actin (gray). (C) Representative confocal images of IHCs from the apical turn of Clic5−/− injected with ssAAV.Clic5 at 12 weeks, stained with a CLIC5 (red), actin (gray), and FLAG tag (green). (D) Representative confocal images of IHCs from the apical turn of Clic5+/+, Clic5−/−, and Clic5−/− injected with ssAAV.Clic5 at 8 weeks, stained with a RDX (magenta) and actin (gray). (E) Representative confocal images of IHCs from the apical turn of Clic5+/+, Clic5−/−, and Clic5−/− injected with ssAAV.Clic5 at 8 weeks, stained with a TPRN (magenta) and actin (gray). Data information: Scale bars = 10 μm for (A), 2 μm for (B–E). Source data are available online for this figure.

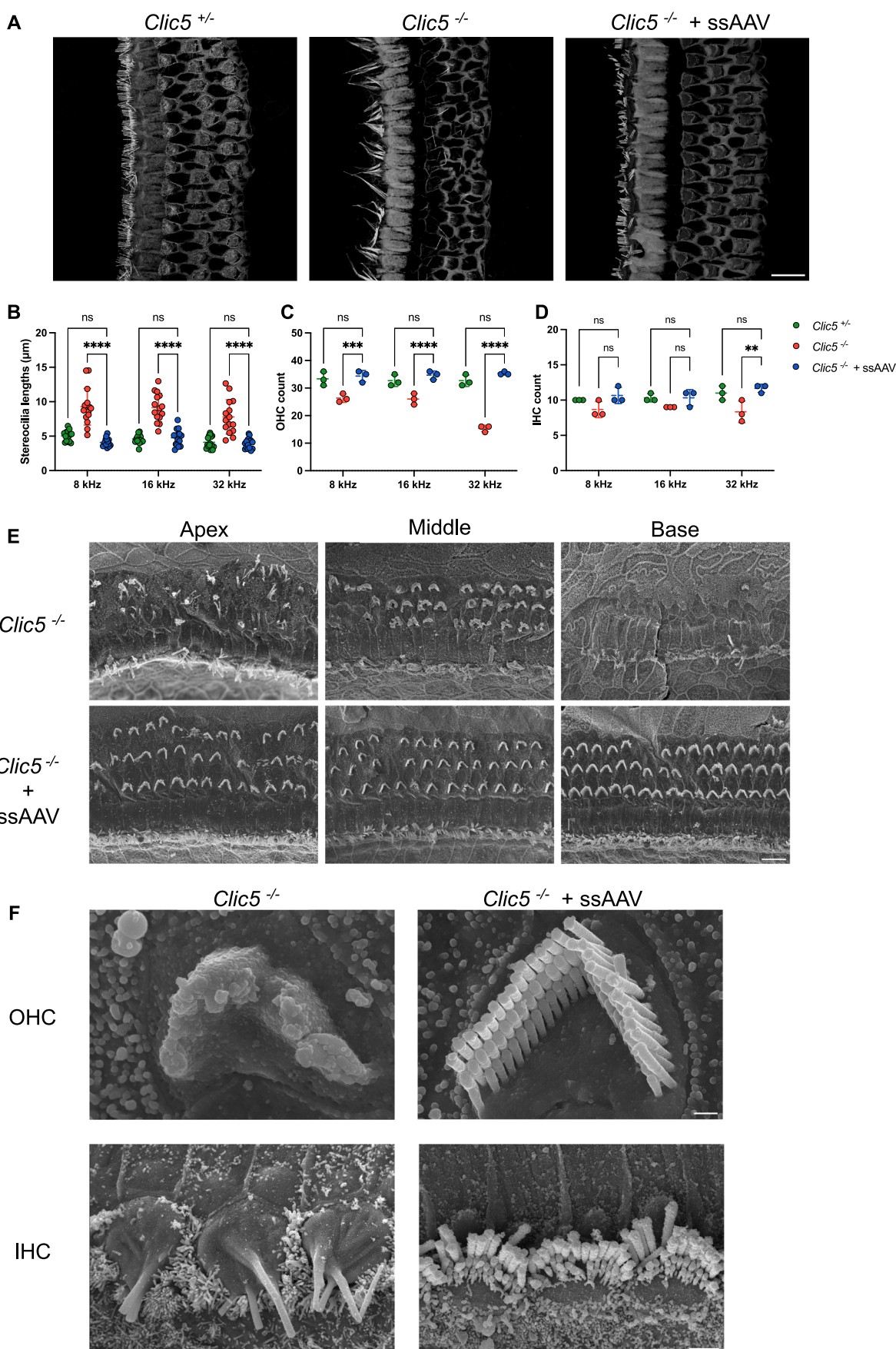

**Figure 3. Single-stranded AAV restores morphology in the auditory system.**

(A) Representative confocal images of IHCs and OHCs from the cochlear 8 kHz region of $Clic5^{+/-}$, $Clic5^{-/-}$ and $Clic5^{-/-}$ injected with ssAAV.$Clic5$ at 12 weeks, stained with actin. (B) Stereocilia length measurements of IHC at 12 weeks at different regions of the cochlea, $n = 3$ for $Clic5^{+/-}$, $Clic5^{-/-}$ and $Clic5^{-/-}$ injected with ssAAV.$Clic5$. (C) Quantification of OHC survival at different regions of the cochlea at 12 weeks, $n = 3$ for $Clic5^{+/-}$, $Clic5^{-/-}$ and $Clic5^{-/-}$ injected with ssAAV.$Clic5$. (D) Quantification of IHC survival at different regions of the cochlea at 12 weeks, $n = 3$ for $Clic5^{+/-}$, $Clic5^{-/-}$ and $Clic5^{-/-}$ injected with ssAAV.$Clic5$. (E) Representative scanning electron microscopy images of IHCs and OHCs at 12 weeks at different regions of the cochlea of $Clic5^{-/-}$ and $Clic5^{-/-}$ injected with ssAAV.$Clic5$. (F) High-magnification images of outer and inner hair bundles from the cochlear 32 kHz region of the $Clic5^{-/-}$ and $Clic5^{-/-}$ injected with ssAAV.$Clic5$ at 12 weeks. Data information: The statistical test was two-way ANOVA followed by Holm–Sidak correction for multiple comparisons. Plots show mean ± SD. ns not significant, **$P < 0.01$, ***$P < 0.001$, ****$P < 0.0001$ (Exact $P$ values are provided in Appendix Table S1). Scale bars = 10 μm for (A, E), 0.5 μm for OHC (F), and 2 μm for IHC (F). Source data are available online for this figure.

possible toxicity in the vestibular system (Fig. EV2E–G). $Clic5^{-/-}$ mice also demonstrated higher activity levels in the open-field test, as shown by a greater total distance traveled than the control group (Fig. 5H,I). Additionally, these mutant mice displayed circling behavior, evident in their travel routes and the quantification of full-body rotations (Fig. 5H,J). Mice treated with ssAAV.$Clic5$ exhibited reduced hyperactivity, reflected in significantly shorter travel distances and minimal full-body rotations, similar to the control group (Fig. 5I,J). In contrast, mutant mice treated with the control AAV encoding GFP did not show any rescue of the vestibular phenotype, indicating that the rescue was only achieved with an AAV encoding CLIC5 (Fig. EV3A–D).

The time spent in the center, typically associated with lower anxiety or more exploratory behavior, (Ruan and Yao, 2020; Seibenhener and Wooten, 2015) was similar in the mutant mice compared to other groups (Fig. 5K). To differentiate anxiety level from sensory impairment that might affect center time, we measured the frequency of center entries and calculated the ratio of total time to entry frequency. The results indicated that untreated mutant mice exhibited more frequent movement in and out of the center, suggesting that vestibular dysfunction, rather than anxiety, influenced their behavior (Fig. 5L). In contrast, the results of $Clic5^{-/-}$ injected mice were similar to those of the control group. Finally, to evaluate the long-term stability of vestibular function following the treatment, we analyzed each group's travel distance and circling behavior over 3 time points (4, 8, and 12 weeks). Treated mutant mice showed stability in their travel distance for up to 12 weeks, whereas untreated mice maintained high levels (Fig. 5M). The circling behavior in treated mice remained stable, while the phenotype in homozygous $Clic5^{-/-}$ mice progressively deteriorated, showing a consistent increase in full-body rotations (Fig. 5N). Altogether, the data demonstrated rescue of vestibular function in treated mice, reaching scores comparable to those of the control group, suggesting restoration of vestibular function.

## Self-complementary vector demonstrated higher expression efficiency in hair cells

In order to compare ssAAV and scAAV expression efficiency, $Clic5^{+/+}$ neonatal mice were injected at P0 with either ssAAV.GFP or scAAV.GFP using the utricle approach for inner ear delivery. Vectors were injected separately in equal volumes of 1.2 μl and equal titer of $3.5 \times 10^9$ gc. Two weeks after injections, the inner ears were harvested, stained with myosin VIIa, and imaged to detect GFP expression. Images were taken under the same conditions and compared with uninjected controls to exclude autofluorescence. Positive GFP-expressing cells were quantified as a percentage of myosin VIIa-labeled cells, with counts obtained from three injected mice per group. Both AAVs demonstrated high expression

efficiency in utricular VHC, with GFP-positive cells in over 97% of VHCs ($P = 0.4453$) (Fig. 6A,B). Robust expression was also observed in both outer and inner HCs, with over 88% of the HCs expressing GFP (Fig. 6C–E). Higher expression efficiency was observed in IHCs from mice treated with the scAAV, with 99.5% of hair cells expressing GFP compared to 93.5% in mice treated with the ssAAV ($P = 0.0226$). For OHCs, a similar percentage of GFP-positive cells was observed in both groups, with values above 97%.

Next, to assess the fluorescence intensity of each AAV, consistent image acquisition parameters and analysis settings were applied to calculate the mean fluorescence intensity for each sample. The mean GFP fluorescence intensity was significantly higher in mice injected with scAAV compared to those injected with ssAAV for both OHCs and IHCs (Fig. 6F,G; $P < 0.0001$). Among the GFP-positive OHCs, the mean fluorescence intensity of scAAV was approximately three times greater than that of ssAAV (28.14 a.u. for ssAAV vs. 83.60 a.u. for scAAV) (Fig. 6F). The difference was even more pronounced in IHCs, where scAAV showed a nearly 4.4-fold higher intensity (18.29 a.u. for ssAAV vs. 81.32 a.u. for scAAV) (Fig. 6G).

## Self-complementary vector rescues $Clic5$ auditory phenotype

To evaluate the therapeutic efficacy of scAAV, we created a synthetic self-complementary vector with the same components as the ssAAV, except the turboGFP fluorescent reporter, which was omitted due to the vector's 2.4 kb size constraint. To evaluate treatment efficiency, we used a titer of $1.52 \times 10^{13}$ gc/ml, which was an order of magnitude lower than the ssAAV titer. The injection process was under the same conditions as those used for ssAAV.$Clic5$, and a volume of 1.2 μl was injected into the ears of P0 mice. CLIC5 expression was detected in both OHCs and IHCs of the $Clic5^{-/-}$ treated mice, specifically at the proximal region of the hair bundle, and co-localized with the FLAG tag expression (Fig. 7A,B), consistent with observations from the ssAAV treatment (Fig. 2C). The stereocilia of the IHCs were well-organized and of uniform length when stained with phalloidin and assessed 12 weeks after the injection, showing a significant improvement compared to the untreated control group (Fig. 7C,D; $P < 0.0001$). The delivery of scAAV.$Clic5$ also improved HC survival, showing significant differences in the number of OHCs in the 8, 16, and 32 kHz regions and a higher number of surviving IHCs in the basal region (Fig. 7E,F; $P \le 0.0005$, $P = 0.007$, respectively).

Similar observations were made using SEM, where the V-shaped arrangement of the hair cells was evident, with hair bundles appearing separated and organized, in contrast to the fused stereocilia observed in the untreated homozygous mice (Fig. 7G,H). When mice were tested for ABR and DPOAE to assess whether the results matched the morphological rescue, auditory function restoration was observed at all three time points (4, 8, and 12

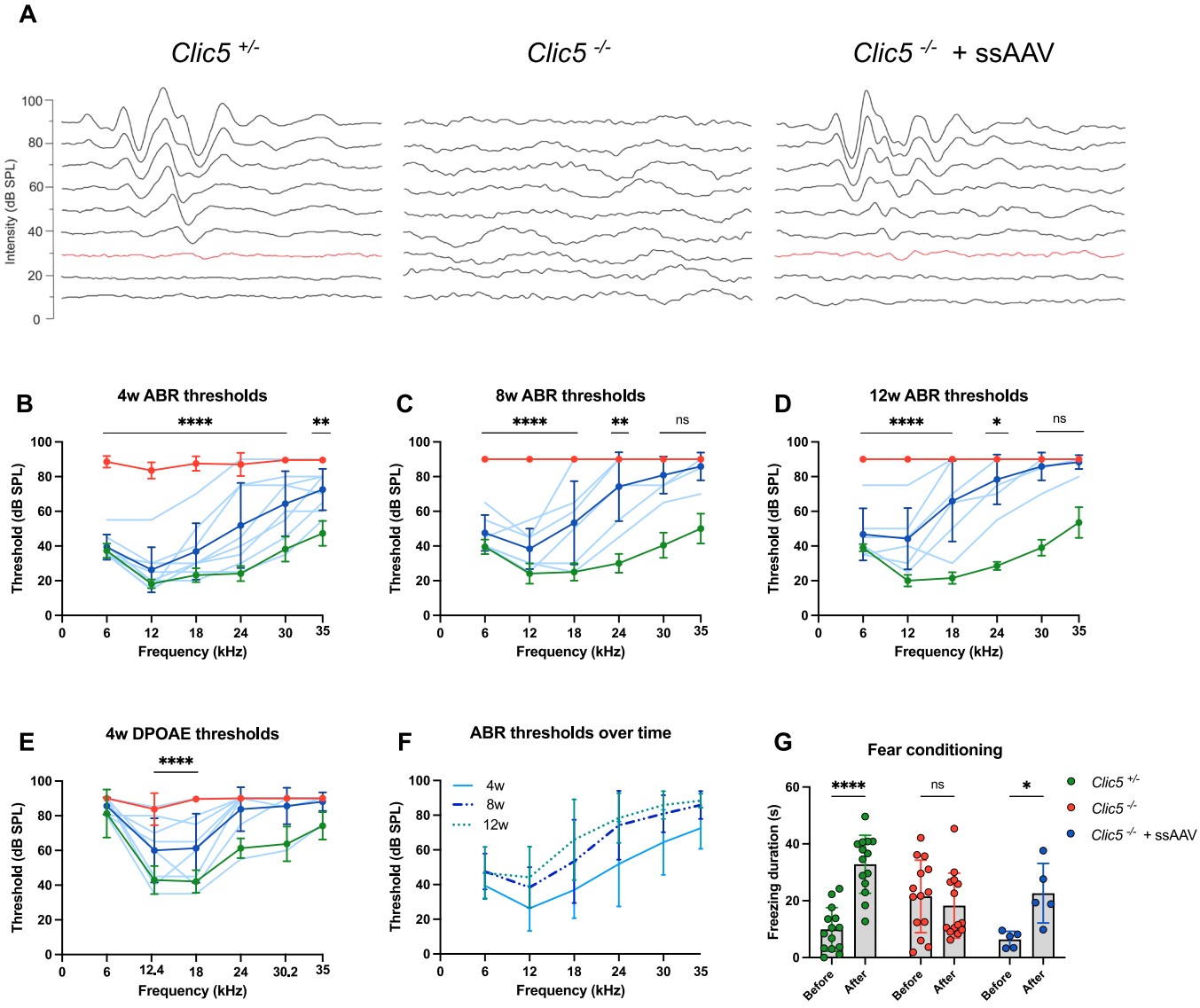

Figure 4. Single-stranded AAV rescues auditory function in *Clic5*⁻/⁻ mice.

(A) Representative ABR recordings obtained at 8 weeks of *Clic5*⁺/⁻, *Clic5*⁻/⁻ and *Clic5*⁻/⁻ injected with ssAAV.*Clic5* in response to 18 kHz stimuli. (B) ABR thresholds at 4 weeks of *Clic5*⁺/⁻ (*n* = 11), *Clic5*⁻/⁻ (*n* = 10), and *Clic5*⁻/⁻ injected with ssAAV.*Clic5* (*n* = 8). (C) ABR thresholds at 8 weeks of *Clic5*⁺/⁻ (*n* = 11), *Clic5*⁻/⁻ (*n* = 11), and *Clic5*⁻/⁻ injected with ssAVV.*Clic5* (*n* = 6). (D) ABR thresholds at 12 weeks of *Clic5*⁺/⁻ (*n* = 10), *Clic5*⁻/⁻ (*n* = 10), and *Clic5*⁻/⁻ injected with ssAAV.*Clic5* (*n* = 6). (E) DPOAE thresholds at 4 weeks of *Clic5*⁺/⁻ (*n* = 12), *Clic5*⁻/⁻ (*n* = 12), and *Clic5*⁻/⁻ injected with ssAVV.*Clic5* (*n* = 8). (F) ABR thresholds at 3-time points of *Clic5*⁻/⁻ injected with ssAAV.*Clic5*: 4 weeks (*n* = 8), 8 weeks (*n* = 6), and 12 weeks (*n* = 6). (G) Freezing behavior duration in a fear conditioning assay at 12 weeks of *Clic5*⁺/⁻ (*n* = 14), *Clic5*⁻/⁻ (*n* = 14), and *Clic5*⁻/⁻ injected with ssAVV.*Clic5* (*n* = 5). Data information: Statistical tests were two-way ANOVA with Holm–Sidak correction for multiple comparisons. Plots show mean ± SD. ns not significant, *$P < 0.05$, **$P < 0.01$, ****$P < 0.0001$ (Exact $P$ values are provided in Appendix Table S1). Source data are available online for this figure.

weeks) (Fig. 7I–L). Hearing sensitivity remained consistent at 4 and 8 weeks but decreased slightly by 12 weeks (Fig. 7M). No changes were observed in the ABRs or DPOAE thresholds of mutant mice injected with scAAV encoding GFP (Fig. EV1B,C). In the fear conditioning assay at 12 weeks, the results aligned with those from the ABR tests, as treated mice exhibited increased freezing behavior in response to the tone (Fig. 7K,N). When comparing auditory sensitivity in *Clic5*⁻/⁻ mice treated with either ssAAV or scAAV, the ssAAV group exhibited a more favorable therapeutic trend only at the 24 and 30 kHz frequencies in the 4-week ABR ($P = 0.0352$

$P = 0.0213$, respectively) (Fig. EV4A). No significant differences in hearing rescue were noted between the two treatment groups for DPOAE results taken at 4 weeks or ABR measurements performed at 8 and 12 weeks (Fig. EV4B–D).

To evaluate the minimum effective dose of scAAV.*Clic5*, inner ear injections were performed as previously described, in *Clic5*⁻/⁻ mice using three different viral titers: the original concentration ($1.52 \times 10^{13}$ gc/ml) and two lower doses ($1.52 \times 10^{12}$ and $1.52 \times 10^{11}$ gc/ml). Four weeks post-injection, auditory function was assessed, and the inner ear was collected for immunostaining.

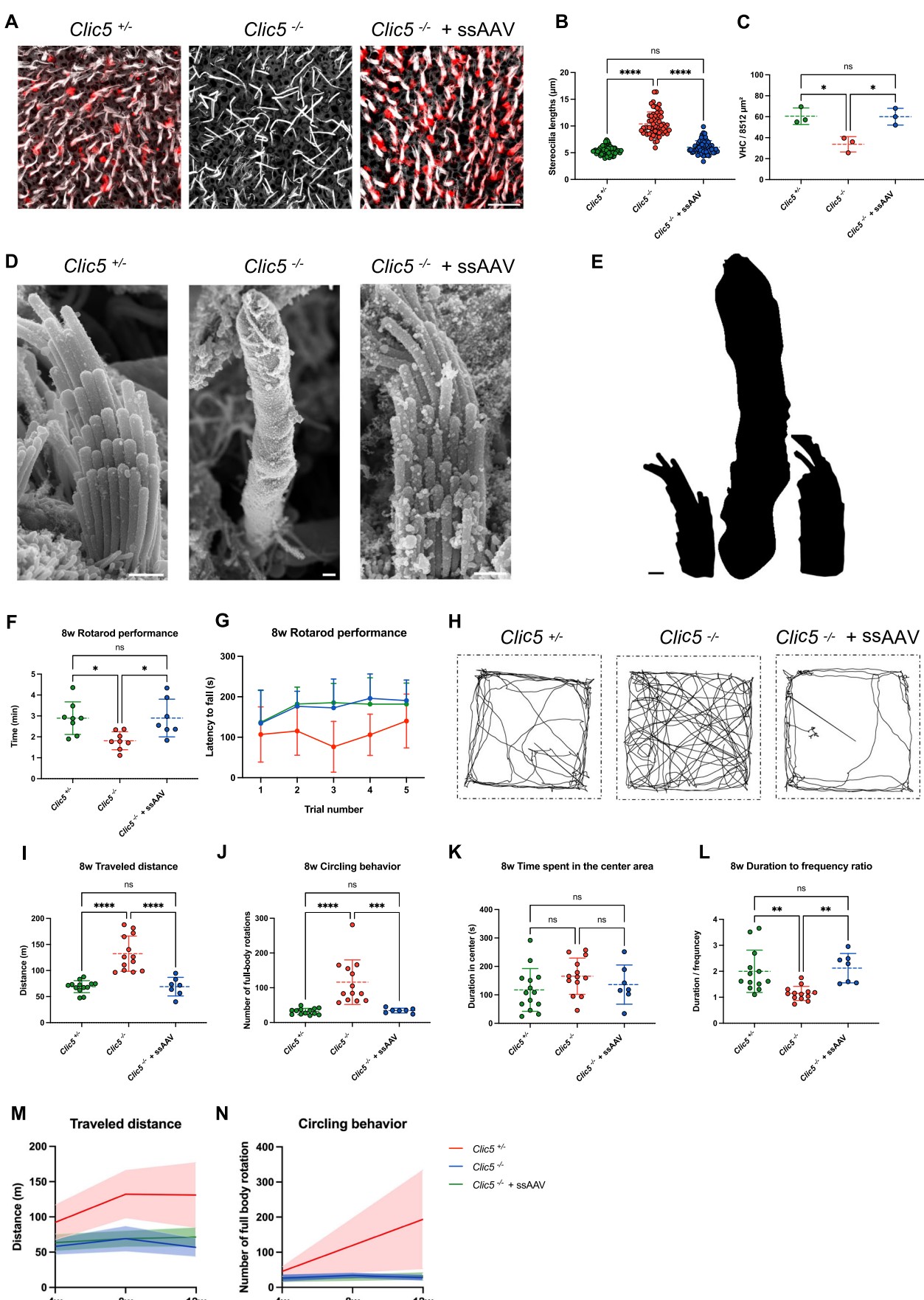

**Figure 5. Single-stranded AAV rescues vestibular phenotype in Clic5⁻/⁻ mice.**

(A) Representative confocal images of VHCs from the utricle of *Clic5⁺/⁻*, *Clic5⁻/⁻* and *Clic5⁻/⁻* injected with ssAAV.*Clic5* at 12 weeks, stained with CLIC5 (red) and actin (gray). (B) Stereocilia length measurements of VHC of the utricle, n = 3 for *Clic5⁺/⁻*, *Clic5⁻/⁻*, and *Clic5⁻/⁻* injected with ssAAV.*Clic5*. (C) Quantification of VHC survival in mouse utricles at 12 weeks, n = 3 for *Clic5⁺/⁻*, *Clic5⁻/⁻*, and *Clic5⁻/⁻* injected with ssAAV.*Clic5*. (D) Representative scanning electron microscopy images of VHCs from the utricle of *Clic5⁺/⁻*, *Clic5⁻/⁻*, and *Clic5⁻/⁻* injected with ssAAV.*Clic5* at 12 weeks. (E) Illustration of hair bundle size differences observed in SEM images. (F) Quantification of the average duration on the rotarod of *Clic5⁺/⁻* (n = 8), *Clic5⁻/⁻* (n = 8), and *Clic5⁻/⁻* injected with ssAAV.*Clic5* (n = 7), performed at 8 weeks. (G) Latency to fall during 5 Trials for *Clic5⁺/⁻* (n = 8), *Clic5⁻/⁻* (n = 8), and *Clic5⁻/⁻* injected with ssAAV.*Clic5* (n = 7), performed at 8 weeks. (H) Representative traveling routes during the 2-min open field of *Clic5⁺/⁻*, *Clic5⁻/⁻*, and *Clic5⁻/⁻* injected with ssAAV.*Clic5*, performed at 8 weeks. (I) Distance traveled in the open-field test of *Clic5⁺/⁻* (n = 13), *Clic5⁻/⁻* (n = 13), and *Clic5⁻/⁻* injected with ssAAV.*Clic5* (n = 7), performed at 8 weeks. (J) Quantification of circling behavior during the open field test of *Clic5⁺/⁻* (n = 13), *Clic5⁻/⁻* (n = 13), and *Clic5⁻/⁻* injected with ssAAV.*Clic5* (n = 7), performed at 8 weeks. (K) Time spent in the center in the open-field test of *Clic5⁺/⁻* (n = 13), *Clic5⁻/⁻* (n = 13), and *Clic5⁻/⁻* injected with ssAAV.*Clic5* (n = 7), performed at 8 weeks. (L) Duration to frequency ratio in the center measurement of an open-field test of *Clic5⁺/⁻* (n = 13), *Clic5⁻/⁻* (n = 13), and *Clic5⁻/⁻* injected with ssAAV.*Clic5* (n = 7), performed at 8 weeks. (M) Averaged distance traveled in the open-field test at 3-time points of *Clic5⁺/⁻*, *Clic5⁻/⁻*, and *Clic5⁻/⁻* injected with ssAAV.*Clic5*. (N) Averaged circling behavior during the open-field test at 3-time points of *Clic5⁺/⁻*, *Clic5⁻/⁻*, and *Clic5⁻/⁻* injected with ssAAV.*Clic5*. Data information: The statistical test was one-way ANOVA followed by Tukey correction for multiple comparisons for (B, C, F, I–L), and two-way ANOVA with Holm–Sidak correction for multiple comparisons for (G, M, N). Plots show mean ± SD. ns = not significant, *P < 0.05, **P < 0.01, ***P < 0.001, ****P < 0.0001 (Exact P values are provided in Appendix Table S1). Scale bars = 10 μm for (A), 1 μm for (D, E). Source data are available online for this figure.

Due to the size limitations of scAAV, a Flag tag was used to track expression instead of GFP. Expression efficiency was quantified by calculating the percentage of Flag-positive cells among myosin VIIa-labeled hair cells. The number of positive hair cells in treated mice increased as a function of the titer (Fig. 7O). When tested for ABR, mice that received the highest titer ($1.52 \times 10^{13}$ gc/ml) exhibited improved hearing sensitivity across a broader range of frequencies (Fig. 7P). The two lower doses resulted in similar thresholds, with rescue primarily limited to low frequencies. Taken together, rescue by scAAV.*Clic5* seems to be dose-dependent.

## Self-complementary vector rescues *Clic5* vestibular phenotype

We next examined the therapeutic effect of the scAAV on the vestibular system. Staining of utricular hair cells revealed robust CLIC5 expression in the hair cells of the treated *Clic5⁻/⁻* mice, along with consistent hair bundle lengths and improved VHC survival, similar to the results observed with ssAAV treatment (Fig. 8A–C). Additionally, SEM analysis showed that the morphology of the utricular hair cells was normal, with the stereocilia arranged in a staircase-like pattern (Fig. 8D). Similar to the hearing rescue observed in mice treated with scAAV.*Clic5*, the morphological improvements correlated with the functional assessment of vestibular function, as treated mice showed improved balance during the rotarod test and demonstrated better performance in the open field test (Fig. 8E–K). No effect was observed when mutant mice received control scAAV encoding GFP (Fig. EV3A–SD). In addition, scAAV treatment had a long-term effect on vestibular function, as evidenced by the maintained travel distance and stable circling behavior observed for up to 12 weeks (Fig. 8L,M). No significant performance differences were observed when comparing the vestibular function of *Clic5⁻/⁻* mice treated with either ssAAV or scAAV across the different assays (Fig. EV5A–C; $P \geq 0.8695$).

## Discussion

AAV vectors have become a leading inner ear gene therapy platform, with recent advancements progressing from preclinical research to clinical trials

(Duhon et al, 2024). Here, we demonstrate that gene replacement therapy, using the utricle injection method, is a viable approach for treating hearing and vestibular disorders associated with *Clic5* mutations. We show the successful restoration of CLIC5 expression in hair cells of the auditory and vestibular sensory systems in *Clic5*-deficient mice, with robust expression observed in the proximal region of the hair bundle. This restoration prevents the fusion and elongation of stereocilia and improves hair cell survival. Beyond the morphological changes caused by the absence of CLIC5, its deficiency has been shown to disrupt the localization of proteins such as RDX, TPRN, and PTPRQ, which are linked to hearing loss (Qi et al, 2024b; Salles et al, 2014). Restoring CLIC5 expression also appears to normalize the localization of RDX, emphasizing CLIC5's vital role in hair cell function. The exogenous delivery of *Clic5* also results in preserving normal hearing and vestibular function, indicated by hearing assessment and various behavioral assays. Vestibular function rescue remained durable up to 12 weeks post-injection, while ABR thresholds deteriorated over time, particularly at high frequencies. The difference between the two systems may result from the fact that VHCs differentiate postnatally in mice and may respond more effectively to therapy, or from differences in stereocilia composition and protein turnover (Das and Manor, 2024; Hastings and Brigande, 2020; Krey and Barr-Gillespie, 2019; McGrath et al, 2017; Zhang et al, 2012). Limited rescue and durability at high frequencies may result from several factors, including promoter downregulation, an immune response against the viral capsid, the earlier development of the cochlear base compared to other cochlear regions, or the genetic background of the mice, including the *Cdh23* allele in the C57BL/6 strain, which is linked to age-related high-frequency hearing loss (Ivanchenko et al, 2021; Patel et al, 2024; Peters et al, 2023). Despite the deterioration of ABR thresholds over time, transgene expression remained robust at 12 weeks, and no significant morphological changes in hair cells were observed.

The ability to rescue auditory and vestibular function in *Clic5*-deficient mice may indicate potential applicability to humans. Hearing loss in individuals with pathogenic variants in the *CLIC5* gene is not congenital but emerges in early childhood, before the second decade of life, while vestibular areflexia, if present, generally develops later (Seco et al, 2015; Wonkam-Tingang et al, 2020). In this study, we performed utricle injections into P0 mice, prior to the morphological changes in hair cells that begin between P14 and P17, at the time of hearing onset. Circling behavior, indicative of vestibular dysfunction, became apparent around 4 weeks of age and was more pronounced by 8 weeks. The effectiveness of both ssAAV and scAAV treatments may indicate a

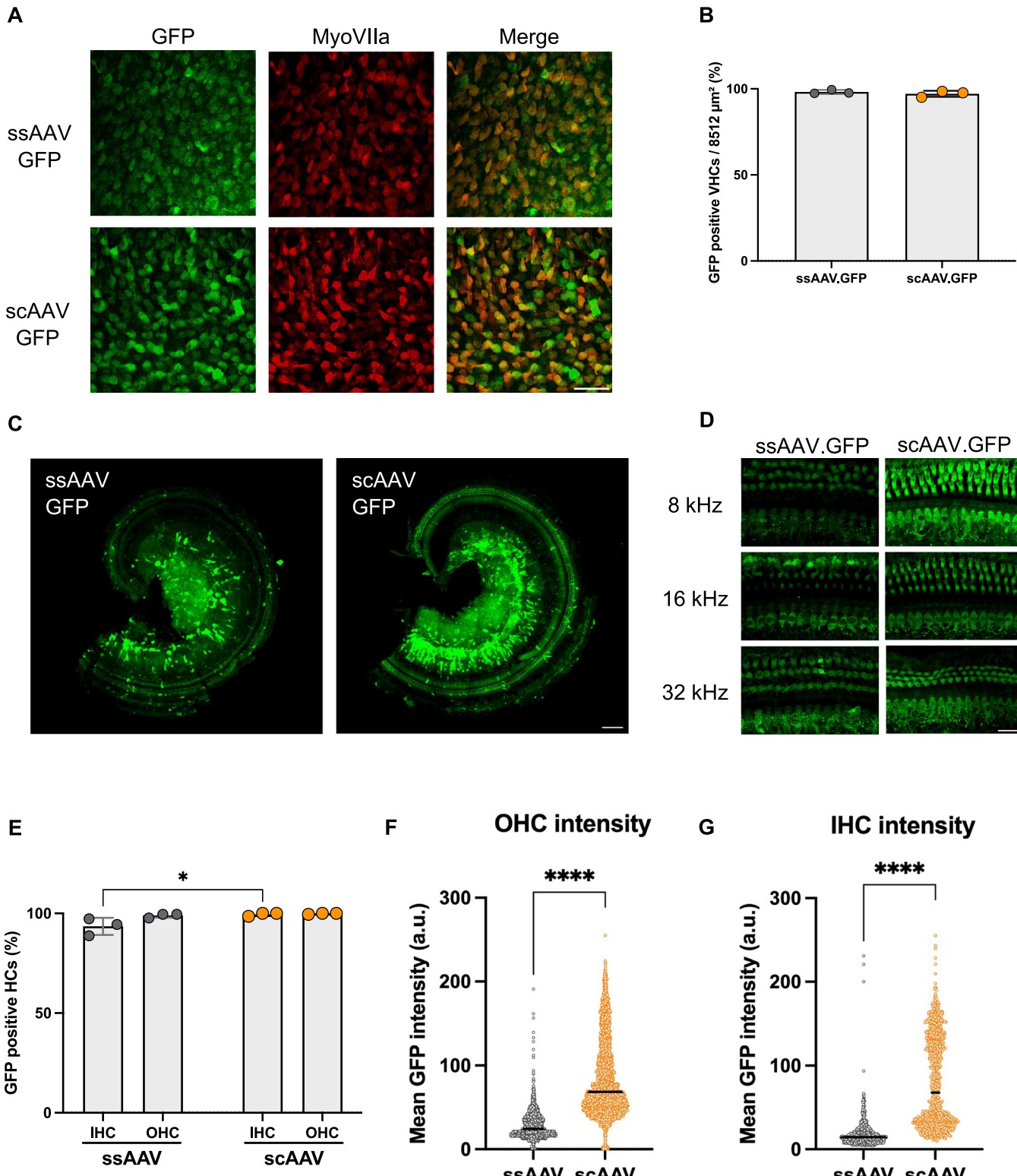

potentially broader therapeutic window for scAAV. While transgene expression with scAAV occurs rapidly, observed within a few days after administration, ssAAV expression typically takes longer (Andino et al, 2007; Ku et al, 2011; Lee et al, 2020). Taken together, we believe that

*CLIC5* gene therapy may be feasible in humans, as hearing loss in these individuals develops postnatally and progresses more gradually than that observed in *Clic5*$^{-/-}$ mice, potentially offering an expanded window of opportunity for treatment.

**Figure 6. Comparison of single-stranded and self-complementary AAVs at equivalent titers.**

(A) High-magnification images of VHCs from the utricle of $Clic5^{+/+}$ injected with ssAAV.GFP or scAAV.GFP at P14. (B) Expression rates of ssAAV.GFP and scAAV.GFP across the utricle, $n = 3$ for each group. (C) Representative confocal images of the organ of Corti of a $Clic5^{+/+}$ injected with ssAAV.GFP or scAAV.GFP at P14. (D) High-magnification images of $Clic5^{+/+}$ injected with ssAAV.GFP or scAAV.GFP at different regions of the cochlea (8, 16 and 32 kHz) at P14. (E) Expression rates of ssAAV.GFP and scAAV.GFP across the cochlea, $n = 3$ for each group. (F) Mean GFP intensity quantification of OHC across the cochlea, $n = 3$ for each group. (G) Mean GFP intensity quantification of IHC across the cochlea, $n = 3$ for each group. Data information: Statistical tests were two-way ANOVA with Holm–Sidak correction for multiple comparisons for (E) and unpaired Student's $t$ test for (B, F, G). Plots show mean ± SD. *$P < 0.05$, ****$P < 0.0001$ (Exact $P$ values are provided in Appendix Table S1). Scale bars = 10 μm for (A), 100 μm for (C), 5 μm for (D). Source data are available online for this figure.

Interestingly, our data show that the scAAV vector achieved the same functional rescue as the ssAAV but at a significantly reduced titer-an order of magnitude lower (ssAAV- $1.69 \times 10^{14}$ gc/ml versus scAAV $1.52 \times 10^{13}$ gc/ml). This reduction in dosing requirement suggests a promising therapeutic strategy, as lower vector doses inherently decrease risks such as immune activation and production challenges (Costa Verdera et al, 2020; Hamilton and Wright, 2021; Mingozzi and High, 2013). Although immune responses to AAVs in the inner ear are not fully understood, other clinical trial data suggest that AAV vector immunogenicity is partly dose-dependent. Lower vector doses tend to induce milder, more manageable inflammation without completely losing transgene expression (George et al, 2017; Ishibashi et al, 2023; Mingozzi et al, 2009; Nathwani et al, 2014). Nonetheless, it is important to highlight that self-complementary genomes have the potential to generate more intense immune responses at specific doses than ssAAV vectors (Martino et al, 2011; Wu et al, 2012). Our dose-response analysis showed that even reduced titers of scAAV.Clic5 can mediate partial hearing restoration (at levels as low as $1.52 \times 10^{11}$ gc/ml), supporting the feasibility of lower dosing strategies. Future studies should explore optimal dosing regimens to maximize efficacy while minimizing adverse immune responses, particularly in the context of inner ear applications. Another critical aspect to consider is the need for large-scale production of recombinant AAVs with higher titers needed during clinical phases (Asaad et al, 2023). In this context, using scAAV vectors, which achieve similar therapeutic outcomes, provides an advantage by enhancing the feasibility of scaling up for clinical trial applications.

The success of gene therapy relies on timely intervention, which can differ among various models. The main advantage of scAAV is its rapid expression, making it particularly suitable for models with a narrow intervention time frame. A study conducted in the *NMNAT1* retinal degeneration model demonstrates the potential of scAAV, where its fast activation successfully rescued retinal structure and function, while ssAAV failed to achieve similar outcomes due to the limited time frame for effective treatment (Greenwald et al, 2020). This potential may be similarly applied to treating different progressive forms of hearing loss, where timely intervention is crucial to prevent irreversible damage, such as *TMPRSS3*-related hearing loss or the *EPS8* model, where EPS8 expression could not be rescued after P2 (Abu Rayyan et al, 2020; Bademci et al, 2016; Jeng et al, 2022).

The primary limitation of scAAV is its small genome capacity, restricted to ~2.4 kb. However, comparative analyses have demonstrated that scAAV vectors can accommodate genomes up to 3–3.3 kb (Wu et al, 2007). Furthermore, over 30 genes associated with hearing loss are small enough to fit within the scAAV vector, including some of the most prevalent causes of hearing loss worldwide, such as *GJB2* and *TMPRSS3* (Table EV1). This, combined with its proven potential for treating hearing and vestibular disorders, positions scAAVs as a viable and promising option for gene replacement therapy in these conditions. The ability to use scAAVs for effective treatment addresses current therapeutic gaps and adds to the growing toolbox of gene therapy strategies for a broader range of genetic hearing loss conditions.

## Methods

### Reagents and tools table

| Experimental models | | |
|---|---|---|
| C57BL/6J-$Clic5^{nmf318}$/J | Jackson Laboratory | #005329 |
| **Recombinant DNA** | | |
| ssAAV9/PHP.B-CB6-Clic5-3XFLAG-P2A-TurboGFP-WPRE | Viral Vector Core at Boston Children's Hospital | N/A |
| scAAV9/PHP.B-CB6-Clic5-3XFLAG-P2A-TurboGFP-WPRE | Viral Vector Core at Boston Children's Hospital | N/A |
| ssAAV9/PHP.B-CB6-TurboGFP-WPRE_Control | Viral Vector Core at Boston Children's Hospital | N/A |
| scAAV9/PHP.B-CB6-TurboGFP-WPRE_Control | Viral Vector Core at Boston Children's Hospital | N/A |
| **Antibodies** | | |
| Polyclonal Myosin VIIa | Proteus Biosciences | cat# 25-6790 |
| ANTI-CLIC5 | Alomone Labs | cat# ACL-025 |
| Monoclonal ANTI-FLAG | Sigma | cat# F3165 |
| ANTI-TPRN | Sigma | cat# HPA020899 |
| ANTI-Radixin | Sigma | cat# R3653 |
| Goat Anti-Mouse IgG H&L (Alexa Fluor® 405) | Abcam | cat# ab175660 |
| Goat Anti-Mouse IgG H&L (Alexa Fluor® 647) | Abcam | cat# ab150119 |
| Goat Anti-Rabbit IgG H&L (Alexa Fluor® 488) | Abcam | cat# ab150081 |

| | | |
|---|---|---|
| Goat Anti-Rabbit IgG H&L (Alexa Fluor® 594) | Abcam | cat# ab150080 |
| Phalloidin-iFluor 647 | Abcam | cat# ab176759 |
| Phalloidin-iFluor 488 | Abcam | cat# ab176753 |
| **Oligonucleotides and other sequence-based reagents** | | |
| Clic5_FWD | | GATATGCACAGCCACCCATG |
| Clic5_REV | | AGGCAATGGGGAAGAAGAGT |
| **Chemicals, enzymes and other reagents** | | |
| HS Taq Mix Red | PCRBIO | cat# PB10.23 |
| EPPic Fast Kit | A&A Biotechnology | cat#1021-500F |
| EDTA 0.5 M, PH 8.0 Sterile Biotechnology Grade | Bio-Lab | cat#009012233100 |
| Paraformaldehyde 16% Aqueous Solution EM Grade | Electron Microscopy Sciences | cat# 15710 |
| Glutaraldehyde 8% solution | Electron Microscopy Sciences | cat# 16020 |
| Osmium tetroxide 4% | Electron Microscopy Sciences | cat# 19170 |
| Thiocarbohydrazide | Sigma | cat# 223220 |
| Triton X-100 | Sigma | cat# 9036-19-5 |
| Normal Goat Serum | Abcam | cat# ab7481 |
| Phosphate Green antibody diluent | Bar Naor Ltd | cat# 2663 |
| Fluorescence Mounting Media | ORIGENE | cat# SKU E18-18 |
| **Software** | | |
| Ethovision XT | https://noldus.com/ethovision-xt | |
| Graphpad prism | https://www.graphpad.com/ | version 10.4.2 |
| ImageJ | https://imagej.net/ij/ | |
| BioSigRZ software | Tucker-Davis Technologies, Alachua, FL | |
| R algorithm | RStudio, Boston, MA | |
| Imaris 3D Interactive Microscopy Visualization software | Oxford Instruments. | version 10.1 |
| Adobe Illustrator | Adobe Systems | |
| SnapGene | https://www.snapgene.com/ | version 8.1 |
| **Other** | | |
| Leica SP8 Confocal microscope | Leica | |
| Zeiss GeminiSEM 300 high-resolution field emission scanning electron microscope | Zeiss | |
| Critical point dryer | Balzer | |

## Mice

The C57BL/6J-*Clic5*[nmf318]/J mouse strain was obtained from the Jackson Laboratory, stock #005329 (Bar Harbor, ME). Mice were kept in a temperature-controlled environment with a 12-h light-dark cycle. Genotyping was performed using HS Taq Mix Red (PCRBIO, PB10.23) with the primers Clic5_FWD: GATATGCA-CAGCCACCCATG and Clic5_REV: AGGCAATGGGGAAGAA-GAGT. The cycling conditions consisted of an initial 3-min denaturation at 95 °C, followed by 35 cycles of 15 s at 95 °C, 30 s at 60 °C, and 15 s at 72 °C. A final elongation step was performed for 10 min at 72 °C and a cool-down for 3 min at 12 °C. PCR products were cleaned using the EPPic Fast Kit (A&A Biotechnology, 1021-500 F) and sent for Sanger sequencing to confirm the *Clic5* mutation. Male and female mice were randomly selected for the study.

## AAV generation

AAV plasmids were produced by VectorBuilder carrying the chicken beta-actin-promotor (CB6), mouse *Clic5* coding sequence (NM_172621.2), and the woodchuck hepatitis virus post-transcriptional regulatory element (WPRE). All AAV viral vectors were prepared at the Viral Vector Core at Boston Children's Hospital as described previously, using the AAV9-PHP capsid (Lee et al, 2020). Viral titers were quantified by performing qPCR amplification using primers specific to the AAV2 ITR sequences. The titers were as listed below: ssAAV9/PHP.B -CB6-Clic5-3XFLAG-P2A-TurboGFP-WPRE 1.69E + 14 gc/ml, scAAV9/PHP.B -CB6-Clic5-3XFLAG-P2A-Tur-boGFP-WPRE- 1.52E + 13 gc/ml, ssAAV9/PHP.B -CB6-TurboGFP-WPRE_Control- 6.73E + 13 gc/ml, scAAV9/PHP.B -CB6-TurboGFP-WPRE_Control- 6.2E + 12 gc/ml. Viruses were aliquoted and stored at −80 °C until use.

## In vivo viral injections

Inner ear injections via the utricle were performed in neonatal mice at P0. Mice were anesthetized by a hypothermic ice bath for 4 min. The procedure was performed under an operating binocular while keeping the mice on a cold surface. Under sterile conditions, a small incision was made to identify and utricle visually. 1.2 µl of the virus was injected into the left utricle of the mouse using a glass pipette (Drummond, Broomall, PA 2-000-100), a P-30 vertical micropipette puller (Sutter Instrument, Novato, CA) and a CMA 102 Microdialysis Pump (CMA, Sweden). After the injection, the incision site was closed with a single 8-0 polypropylene suture and iodine cream were applied. Mice were recovered on a heating pad and returned to their home cage. The total surgery time was kept under 15 min.

## Scanning electron microscopy

The temporal bone was dissected out of the mouse after $CO_2$ euthanasia. Dissection was performed in PBS buffer, and holes were created to enhance fixative perfusion in the apical region of the temporal bone and the round and oval window membranes. Samples were fixed in 2.5% glutaraldehyde (Electron Microscopy Sciences) for ~30 min. Temporal bones were rinsed with PBS buffer and then incubated for ~48–72 h in 0.25 M EDTA at 4 °C. Samples were rewashed in PBS, and a microdissection was performed under a

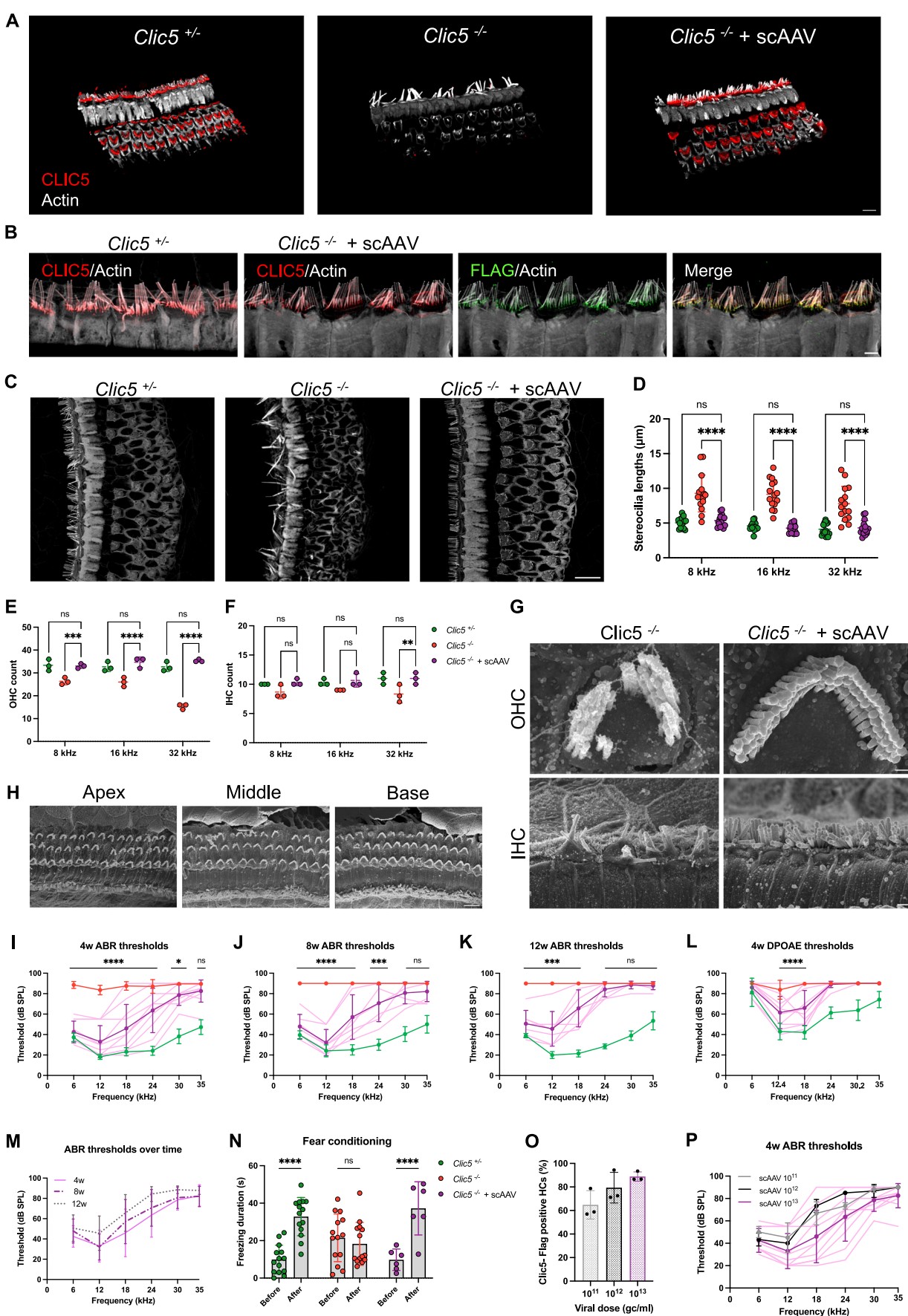

◀ **Figure 7. Self-complementary AAV rescues auditory phenotype in *Clic5*⁻/⁻ mice.**

(A) 3D surface projection of IHCs and OHCs from the cochlear 32 kHz region of *Clic5*⁺/⁻, *Clic5*⁻/⁻ and *Clic5*⁻/⁻ injected with scAAV.*Clic5* at 12 weeks, stained with a CLIC5 (red) and actin (gray). (B) Representative confocal images of IHCs from the apical turn of *Clic5*⁺/⁻ and *Clic5*⁻/⁻ injected with scAAV.*Clic5* at 12 weeks, stained with a CLIC5 (red), actin (gray), and FLAG tag (green). (C) Representative confocal images of IHCs and OHCs from the cochlear 8 kHz region of *Clic5*⁺/⁻, *Clic5*⁻/⁻, and *Clic5*⁻/⁻ injected with scAAV.*Clic5* at 12 weeks, stained with actin. (D) Stereocilia length measurements of IHC at different regions of the cochlea, $n = 3$ for *Clic5*⁺/⁻, *Clic5*⁻/⁻, and *Clic5*⁻/⁻ injected with scAAV.*Clic5*. (E) Quantification of OHC survival at different regions of the cochlea, $n = 3$ for *Clic5*⁺/⁻, *Clic5*⁻/⁻, and *Clic5*⁻/⁻ injected with scAAV.*Clic5*. (F) Quantification of IHC survival at different regions of the cochlea, $n = 3$ for *Clic5*⁺/⁻, *Clic5*⁻/⁻, and *Clic5*⁻/⁻ injected with scAAV.*Clic5*. (G) High-magnification scanning electron microscopy images of outer and inner hair bundles from the cochlea middle-turn of the *Clic5*⁻/⁻ and *Clic5*⁻/⁻ injected with scAAV.*Clic5* at 12 weeks. (H) Representative images of IHCs and OHCs at 12 weeks at different regions of the cochlea of *Clic5*⁻/⁻ injected with scAAV.*Clic5*. (I) ABR thresholds at 4 weeks of *Clic5*⁺/⁻ ($n = 11$), *Clic5*⁻/⁻ ($n = 10$), and *Clic5*⁻/⁻ injected with scAAV.*Clic5* ($n = 10$). (J) ABR thresholds at 8 weeks of *Clic5*⁺/⁻ ($n = 11$), *Clic5*⁻/⁻ ($n = 11$), and *Clic5*⁻/⁻ injected with scAVV.*Clic5* ($n = 7$). (K) ABR thresholds at 12 weeks of *Clic5*⁺/⁻ ($n = 10$), *Clic5*⁻/⁻ ($n = 10$), and *Clic5*⁻/⁻ injected with scAAV.*Clic5* ($n = 7$). (L) DPOAE thresholds at 4 weeks of *Clic5*⁺/⁻ ($n = 12$), *Clic5*⁻/⁻ ($n = 12$), and *Clic5*⁻/⁻ injected with scAAV.*Clic5* ($n = 10$). (M) ABR thresholds at 3-time points of *Clic5*⁻/⁻ injected with scAAV.*Clic5*: 4 weeks ($n = 10$), 8 weeks ($n = 7$), and 12 weeks ($n = 7$). (N) Freezing behavior duration in a fear conditioning assay at 12 weeks of *Clic5*⁺/⁻ ($n = 14$), *Clic5*⁻/⁻ ($n = 14$), and *Clic5*⁻/⁻ injected with scAAV.*Clic5* ($n = 6$). (O) Expression rates of scAAV.*Clic5* across the cochlea as a function of viral titer, $n = 3$ for each group. (P) ABR thresholds at 4 weeks of *Clic5*⁻/⁻ injected with scAAV.*Clic5* at different viral doses: $1.52 \times 10^{13}$ gc/ml ($n = 10$), $1.52 \times 10^{12}$ gc/ml ($n = 3$) and $1.52 \times 10^{11}$ gc/ml ($n = 3$). Data information: Statistical tests were two-way ANOVA with Holm–Sidak correction for multiple comparisons for (D–F, I–N, P), and one-way ANOVA followed by Tukey correction for multiple comparisons for (O). Plots show mean ± SD. ns not significant, $*P < 0.05$, $**P < 0.01$, $***P < 0.001$, $****P < 0.0001$ (Exact $P$ values are provided in Appendix Table S1). Scale bars = 10 μm for (A, C, H), 2 μm for (B), 0.5 μm for OHC (G), and 2 μm for IHC (G). Source data are available online for this figure.

binocular microscope for the cochlear sensory epithelia and vestibular organs. Tissues were treated with alternating incubations in osmium tetroxide and thiocarbohydrazide (OTOTO). Next, tissues were dehydrated using an ethanol gradient from 30 to 70% for 15 min each. Tissues were placed in a critical point dryer (Balzer) at the electron microscopy unit at the Tel Aviv University Faculty of Life Sciences. Finally, samples were mounted on a piece of conductive carbon tape and chrome plated. Images were obtained using Zeiss GeminiSEM 300 high-resolution field emission scanning electron microscope (HRSEM) with 10.0-kV accelerating voltage at the Center for Nanoscience and Nanotechnology at Tel Aviv University.

## Confocal immunofluorescence

Dissection of the inner ear was performed in PBS after neonatal mice were sacrificed by decapitation and adult mice by $CO_2$ euthanasia. Dissection was performed under operating binoculars, creating holes near the oval and round window membranes to allow fixative access. The samples were then fixed in 4% paraformaldehyde (Electron Microscopy Sciences) for 1 h at room temperature and washed three times with PBS. For mice older than P10 samples, decalcification was made using 0.25 M EDTA until the temporal bones were entirely soft. Cochlear sensory epithelia and vestibular organs were microdissected in PBS and placed in a blocking and permeabilization solution of 2% Triton X-100 and 10% normal goat serum for 1 h at room temperature. Tissues were incubated for 2 h at room temperature with the appropriate primary antibody in a dilution of Phosphate Green antibody diluent (Bar Naor Ltd) according to manufacturer instructions. After three washes in PBS, samples were incubated with appropriate secondary antibodies diluted in PBS for 1 h. Following three washes of PBS, samples were placed on carrying glass in Fluorescence Mounting Media (ORIGENE) and covered with a cover glass. Antibodies and stain concentrations were as listed below: rabbit polyclonal myosin VIIa (Proteus Biosciences 25-6790) 1:250, rabbit polyclonal anti-CLIC5 (ACL-025, Alomone labs) 1:100, mouse anti-FLAG (Sigma F3165) 1:1000, rabbit anti-TPRN (Sigma HPA020899) 1:200, rabbit anti-RDX (Sigma R3653) 1:200, goat anti-mouse (Abcam, ab175660) 1:250, goat anti-mouse (Abcam, ab150119) 1:250, goat anti-rabbit (Abcam, ab150080) 1:250, goat anti-rabbit Alexa Fluor 647-phalloidin (Abcam, ab176759) 1:1000, goat anti-rabbit Alexa Fluor 488-phalloidin (Abcam, ab176753) 1:250. Images were taken using

the Leica SP8 Confocal microscope and analyzed using Fiji-ImageJ software and Imaris 10.1 3D Interactive Microscopy Visualization software.

## Hearing assessment

Mice were anesthetized using a combination of ketamine (100 mg/kg) and xylazine (10 mg/kg) by intraperitoneal injection. A heating pad maintained body temperature at 37 °C throughout the experiment. For ABR recordings, three subdermal electrodes were connected to the mouse's head to measure responses. Mice were exposed to click stimuli and pure tones at frequencies of 6, 12, 18, 24, 30, and 35 kHz, with sound intensities ranging from 10 to 90 dB SPL in 5 dB increments. In total, 512 responses were recorded and averaged for each frequency–intensity combination. The lowest sound intensity that elicited a reproducible waveform was defined as the ABR threshold. For DPOAE recording, two speakers, each delivering primary tones (f1 and f2) at a frequency ratio of 1.2. were connected to a microphone and placed within the ear canal of the mice. The results were averaged after each frequency–tone combination was presented 256 times. For DPOAE recording, two speakers presenting two primary tones (f1 and f2) at a frequency ratio of 1.2 were coupled to a microphone and introduced into the ear canal of mice. Each frequency–tone combination was presented 256 times, and the results were averaged. Using a designated R algorithm, the amplitude of the distortion product at a frequency of 2f1–f2, and the surrounding average noise were extracted from the averaged responses. Measurements were performed using an acoustic chamber (MAC-1, Industrial Acoustic Company, Naperville, IL, USA), an RZ6 multiprocessor, an ER-10b + microphone (Etymotic Research, Elk Grove Village, IL), and MF1 speakers (Tucker-Davis Technologies, Alachua, FL). The measurements were analyzed using BioSigRZ software (Tucker-Davis Technologies, Alachua, FL) and a designated R algorithm (RStudio, Boston, MA).

## Open-field test

The open-field test was conducted using a 2.5 m² square arena with overhead white lighting. Mice were placed inside the square arena for

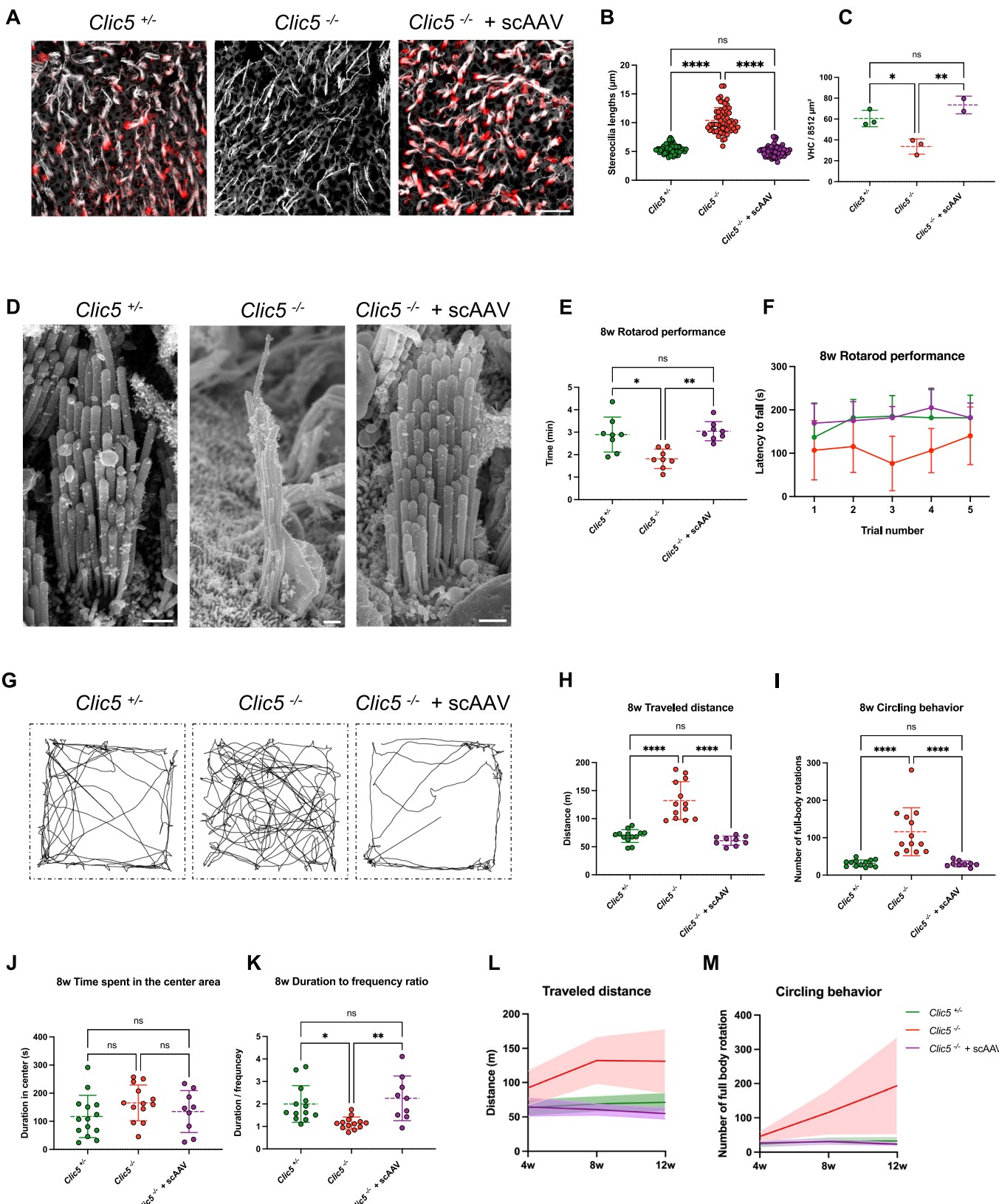

**Figure 8. Self-complementary AAV rescues vestibular phenotype in Clic5⁻/⁻ mice.**

(A) Representative confocal images of VHCs from the utricle of $Clic5^{+/-}$, $Clic5^{-/-}$ and $Clic5^{-/-}$ injected with scAAV.Clic5 at 12 weeks, stained with CLIC5 (red) and actin (gray). (B) Stereocilia length measurements of VHC of the utricle, $n = 3$ for $Clic5^{+/-}$, $Clic5^{-/-}$ and $Clic5^{-/-}$ injected with scAAV.Clic5. (C) Quantification of VHC survival in mouse utricles at 12 weeks, $n = 3$ for $Clic5^{+/-}$ and $Clic5^{-/-}$, $n = 2$ for $Clic5^{-/-}$ injected with scAAV.Clic5. (D) Representative scanning electron microscopy images of VHCs from the utricle of $Clic5^{+/-}$, $Clic5^{-/-}$, and $Clic5^{-/-}$ injected with scAAV.Clic5 at 12 weeks. (E) Quantification of the average duration on the rotarod of $Clic5^{+/-}$ ($n = 8$), $Clic5^{-/-}$ ($n = 8$), and $Clic5^{-/-}$ injected with scAAV.Clic5 ($n = 8$), performed at 8 weeks. (F) Latency to fall during 5 Trials for $Clic5^{+/-}$ ($n = 8$), $Clic5^{-/-}$ ($n = 8$), and $Clic5^{-/-}$ injected with scAAV.Clic5 performed at 8 weeks. (G) Representative traveling routes during the 2-min open field of $Clic5^{+/-}$, $Clic5^{-/-}$, and $Clic5^{-/-}$ injected with scAAV.Clic5 performed at 8 weeks. (H) Distance traveled in the open-field test of $Clic5^{+/-}$ ($n = 13$), $Clic5^{-/-}$ ($n = 13$), and $Clic5^{-/-}$ injected with scAAV.Clic5 ($n = 9$), performed at 8 weeks. (I) Quantification of circling behavior during the open field test of $Clic5^{+/-}$ ($n = 13$), $Clic5^{-/-}$ ($n = 13$), and $Clic5^{-/-}$ injected with scAAV.Clic5 ($n = 9$), performed at 8 weeks. (J) Time spent in the center in the open-field test of $Clic5^{+/-}$ ($n = 13$), $Clic5^{-/-}$ ($n = 13$), and $Clic5^{-/-}$ injected with scAAV.Clic5 ($n = 9$), performed at 8 weeks. (K) Duration to frequency ratio in the center measurement of an open-field test of $Clic5^{+/-}$ ($n = 13$), $Clic5^{-/-}$ ($n = 13$), and $Clic5^{-/-}$ injected with scAAV.Clic5 ($n = 9$), performed at 8 weeks. (L) Averaged distance traveled in the open-field test at 3-time points of $Clic5^{+/-}$, $Clic5^{-/-}$, and $Clic5^{-/-}$ injected with scAAV.Clic5. (M) Averaged circling behavior during the open-field test at 3-time points of $Clic5^{+/-}$, $Clic5^{-/-}$ and $Clic5^{-/-}$ injected with scAAV.Clic5. Data information: The statistical test was one-way ANOVA followed by Tukey correction for multiple comparisons for (B, C, E, H–K), and two-way ANOVA with Holm–Sidak correction for multiple comparisons for (F, L, M). Plots show mean ± SD. ns not significant, *$P < 0.05$, **$P < 0.01$, ***$P < 0.001$ ****$P < 0.0001$ (Exact $P$ values are provided in Appendix Table S1). Scale bars = 10 μm for (A), 1 μm for (D). Source data are available online for this figure.

15 min and behavior was recorded using EthoVision XT software (Noldus), enabling tracking of the center point of the mouse and its nose to detect rotational movements and traveled distance. Data was analyzed after the initial 2 min to allow the mouse to become accustomed to the new environment. Open-field assessments for distance travel patterns were analyzed for 2-min and rotation behavior was calculated as full-body rotations including both clockwise and counterclockwise. All experiments were conducted during the day, and assessments were conducted blindly.

## Rotarod

The rotarod test was performed using a rotating rod with an accelerating speed for 300 s. The starting speed was 5 rpm and increased by 9 rpm/min until a maximum speed of 50 rpm. Mice were placed on the rotating rods, and the duration they could remain on before falling off was measured. A total of five trials were conducted, with a 2-min rest period between each trial. The average of the five trials was calculated.

## Fear conditioning

The Fear conditioning test was conducted in a dedicated acoustic chamber with a sound generator, a shock grid, and a camera. The assay was conducted over two consecutive days, with a learning day followed by a test day. On the training day, mice were placed in the chamber, and an acclimation period of 1.5 min was allowed. After acclimation, a 6 kHz conditioned tone was presented for 20 s, concluding with a mild foot shock of 0.7 mA for 2 s. Following an additional 60-s interval, the tone presented again, followed by a second shock. Twenty-four hours later, the mice were placed back in the same chamber for the test phase. After an initial acclimation period of 2.5 min, the same tone was presented again without the shock, and freezing behavior was recorded for 2.5 min. Behavioral responses were recorded using EthoVision XT software (Noldus) to assess freezing times. The activity was quantified by measuring pixel changes as objective parameters. The "freezing" behavior was measured during two-time intervals: before the tone appeared (01:30–02:30) and after the tone appeared (02:30–03:30).

## Image analysis

Analysis was performed using Fiji-ImageJ software and Imaris 10.1 3D Interactive Microscopy Visualization software. For measurements of the IHC, positions along the cochlea were defined by drawing a polyline starting at the apex. Segments of 92.26 μm with z-stacks were obtained from phalloidin staining to measure the hair bundles of IHC. For each segment, the five longest hair bundles in the center were measured, and for each cochlea, a total of 15 IHCs were measured (8, 16, and 32 kHz). Hair cell survival was assessed using myosin VIIa staining, and counts were performed on the same-sized segments (92.26 μm) from different cochlear regions.

For measurements of the VHC from each sample, two 92.26 μm segments with z-stacks from the striolar region were obtained using phalloidin staining. Two 20 μm squares were placed within the same areas of each segment for all images, and the five longest hair bundles within these regions were measured. A total of 20 VHCs were measured for each utricle. To quantify the number of VHC, hair cells were labeled using myosin VIIa staining. Two images were acquired from consistent regions of each utricle on the same-sized segments (92.26 μm), and myosin VIIa-positive hair cells were counted in each image. The average of the two counts was used as a single biological replicate for each sample. Whole-mount immunofluorescence of the organ of Corti was performed for GFP analysis using a ×63 objective and Z-stack imaging, with consistent conditions for laser settings, z-step size, and pixel ratio. To exclude autofluorescence, uninjected control samples were used. Based on myosin VIIa staining, IHCs and OHCs were identified. The GFP-positive IHCs and OHCs in each sample were quantified separately, and the percentage of GFP-positive cells was calculated by dividing the number of positive cells by the total number of IHCs and OHCs. Similarly, GFP-positive VHCs were quantified using two images per utricle, acquired from consistent regions (92.26 μm segments), and the percentage was calculated relative to the total number of VHCs. Most of the samples were quantified automatically using Imaris software, and the results from manual quantification closely matched those obtained through automation. The GFP mean intensity was measured using the Imaris software from a whole-mount image of the organ of Corti. Plots presenting GFP intensity as arbitrary units.

## The paper explained

### Problem

Hearing loss is genetically diverse, with pathogenic variants in over 150 genes. While gene therapy using adeno-associated virus is a powerful approach for treating inner ear diseases, a gene-specific strategy is necessary to ensure effective therapeutic development. Additionally, refining existing treatment strategies is essential to improving therapeutic outcomes and broadening their applicability. To address this, we investigated *CLIC5*, a gene linked to recessive deafness and vestibular dysfunction in humans and *Clic5*-deficient mice.

### Results

Two synthetic AAV2/9-PHP.B vectors were engineered to deliver a functional copy of *Clic5* into the inner ear of *Clic5*-deficient mice: a single-stranded AAV and a self-complementary AAV, the latter known for rapid transgene expression. The results indicate a decrease in morphological degeneration and restoration of auditory and vestibular function. Moreover, the self-complementary construct achieved similar recovery at a lower titer, offering reduced dose requirements and greater clinical potential for inner ear therapy.

### Impact

These findings have important implications for improving the safety and clinical feasibility of treating hearing and vestibular impairments while emphasizing the potential of self-complementary AAV. This work could pave the way for treating patients with genetic hearing and balance disorders.

## Statistics

Statistical analyses were performed using Prism 10 software (GraphPad version 10.4.2, San Diego, CA). The figure legends detail statistical tests, sample sizes, and P values. Data are presented as mean ± SD. Randomization was used whenever applicable. Data from the untreated control group ($Clic5^{+/-}$ and $Clic5^{-/-}$) for functional assays are identical when compared to either ssAAV treatment or treatment with scAAV.

## Study approval

The procedures involving animals were performed according to the guidelines described in the National Institutes of Health Guide for the Care and Use of Laboratory Animals and were approved by the Animal Care and Use Committees of Tel Aviv University (2211-174-4 and 2211-176-3).

## Data availability

This study includes no data deposited in external repositories.

The source data of this paper are collected in the following database record: biostudies:S-SCDT-10_1038-S44321-025-00275-7.

## Peer review information

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

## Acknowledgements

We thank Lara Kamal and Eyal Marton for their technical assistance and valuable advice. Confocal imaging and image analysis were carried out with the help of Dr. Sasha Lichtenstein at the Research Infrastructure Core Facility (RICF), Gray Faculty of Medical and Health Sciences, Tel Aviv University. Scanning electron microscopy images were conducted with the assistance of Dr. Vered Holdengreber, Adam Cohen, and Gil Daffan at the Center for Nanoscience and Nanotechnology at Tel Aviv University. Behavioral assays were performed at the Myers Neuro-Behavioral Core Facility, Tel Aviv University, under the guidance of Dr. Lior Bikoveski. The research was funded by the United States–Israel Binational Science Foundation (BSF) 01027150 (KBA, GSGG, JRH), the National Institutes of Health/NIDCD R01DC011835 (KBA), and the Breakthrough Award, 2711/22 of the Israel Science Foundation (KBA) Jerusalem, Israel.

## Author contributions

**Roni Hahn**: Conceptualization; Data curation; Formal analysis; Validation; Investigation; Visualization; Methodology; Writing—original draft; Project administration; Writing—review and editing. **Shahar Taiber**: Conceptualization; Investigation; Methodology; Writing—original draft; Writing—review and editing. **Olga Shubina-Oleinik**: Conceptualization; Methodology. **Gwenaëlle SG Géléoc**: Conceptualization; Resources; Supervision; Funding acquisition; Investigation; Methodology; Writing—original draft; Project administration; Writing—review and editing. **Jeffrey R Holt**: Conceptualization; Resources; Supervision; Funding acquisition; Investigation; Methodology; Writing—original draft; Project administration; Writing—review and editing. **Karen B Avraham**: Conceptualization; Resources; Supervision; Funding acquisition; Investigation; Methodology; Writing—original draft; Project administration; Writing—review and editing.

In addition to the CRediT author contributions listed above, the contributions in detail are:

RH, ST, OS-O, GSGG, JRH, and KBA conceived ideas, designed the research, and wrote and edited the paper. OS-O prepared the constructs. RH performed the remainder of the experiments.

Source data underlying figure panels in this paper may have individual authorship assigned. Where available, figure panel/source data authorship is listed in the following database record: biostudies:S-SCDT-10_1038-S44321-025-00275-7.

## Disclosure and competing interests statement

The authors declare no competing interests.

# Expanded View Figures

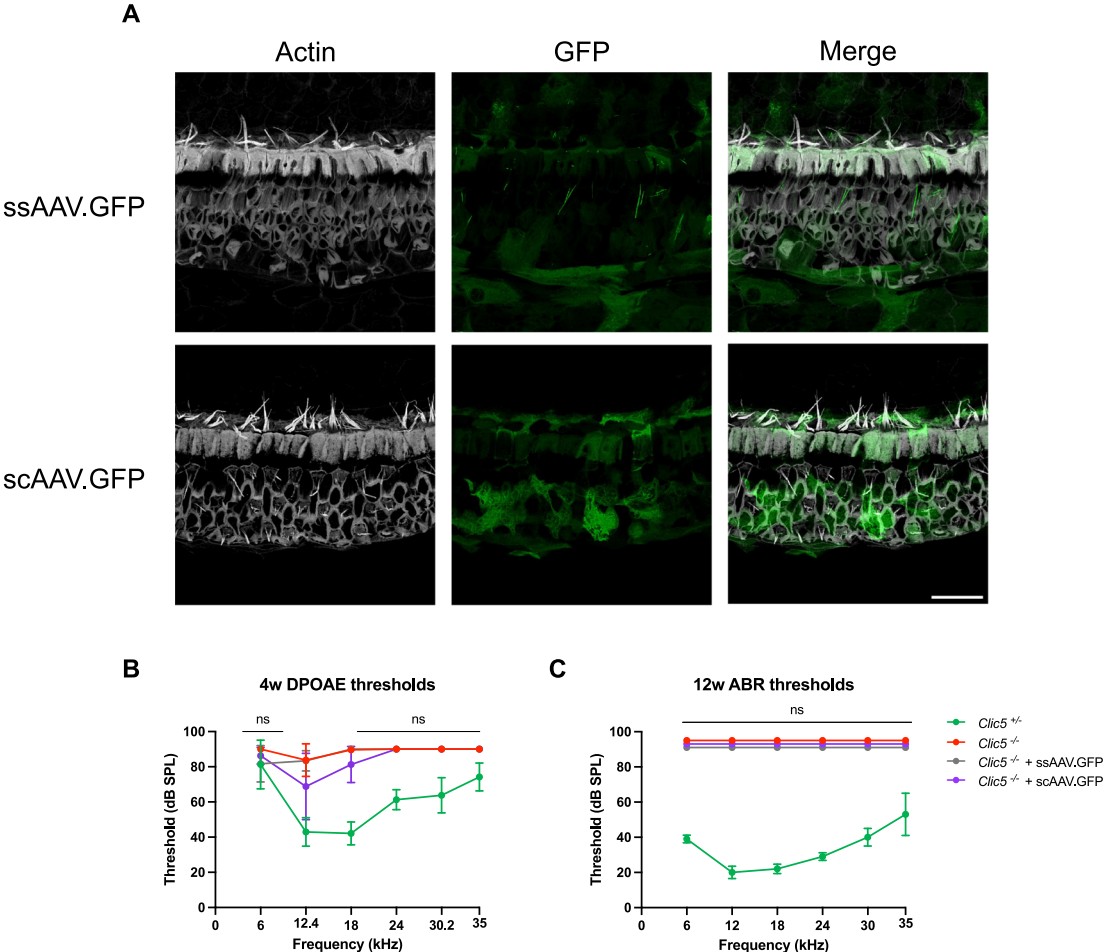

**Figure EV1.   Single-stranded or self-complementary AAV.GFP does not rescue the morphology or function of the auditory system.**

(**A**) Apical region of the cochlea of 12 weeks *Clic5^{−/−}* injected with ssAAV.GFP or scAAV.GFP. (**B**) DPOAE thresholds at 4 weeks of *Clic5^{+/−}* ($n = 12$), *Clic5^{−/−}* ($n = 12$), *Clic5^{−/−}* injected with ssAAV.GFP ($n = 3$) or scAAV.GFP. ($n = 4$). (**C**) ABR thresholds at 12 weeks of *Clic5^{+/−}* ($n = 5$), *Clic5^{−/−}* ($n = 10$), *Clic5^{−/−}* injected with ssAAV.GFP ($n = 3$) or scAAV.GFP. ($n = 4$). Data information: The statistics test was two-way ANOVA with Holm–Sidak correction for multiple comparisons. Plots show mean ± SD. ns = not significant. Scale bar = 10 μm. Source data are available online for this figure.

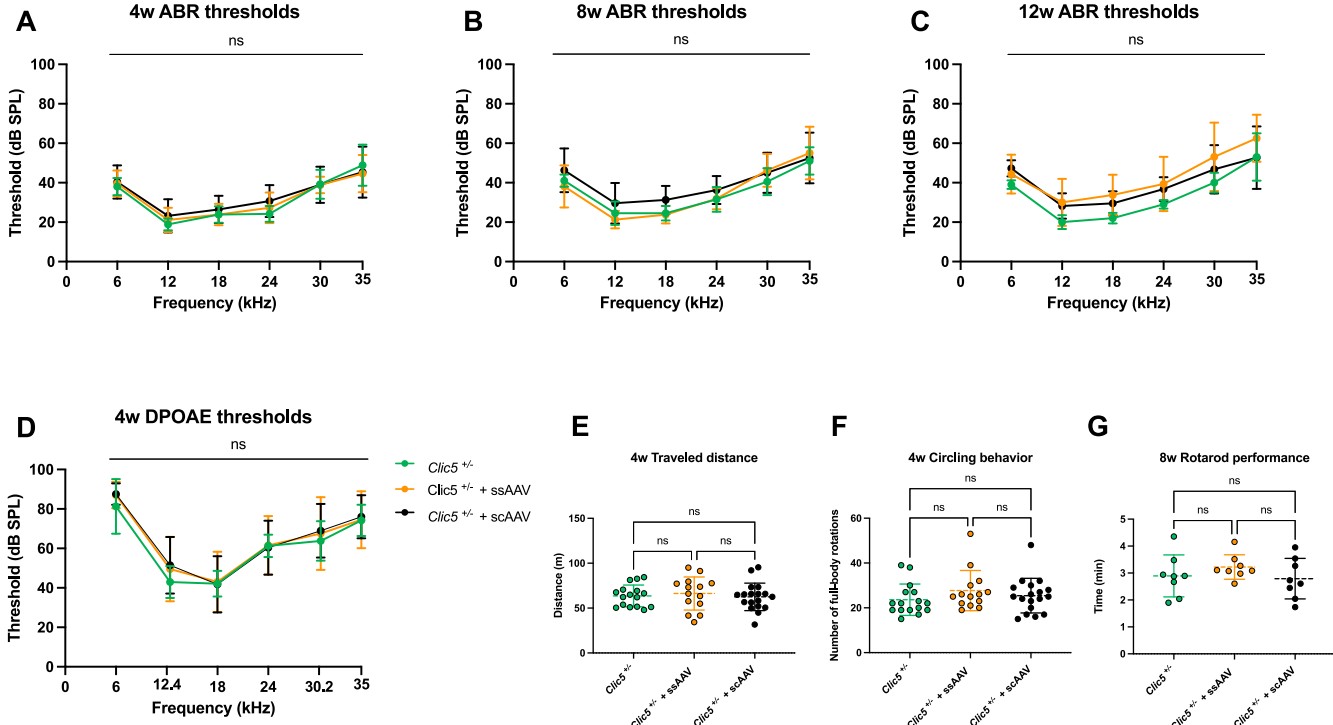

**Figure EV2. Single-stranded or self-complementary AAV.Clic5 does not affect hearing and vestibular function of control mice.**

(A) ABR thresholds at 4 weeks of $Clic5^{+/-}$ ($n = 13$), $Clic5^{+/-}$ injected with ssAAV.Clic5 ($n = 13$) or with scAAV.Clic5 ($n = 14$). (B) ABR thresholds at 8 weeks of $Clic5^{+/-}$ ($n = 10$), $Clic5^{+/-}$ injected with ssAAV.Clic5 ($n = 8$) or with scAAV.Clic5 ($n = 12$). (C) ABR thresholds at 12 weeks of $Clic5^{+/-}$ ($n = 5$), $Clic5^{+/-}$ injected with ssAAV.Clic5 ($n = 8$) or with scAAV.Clic5 ($n = 11$). (D) DPOAE thresholds at 4 weeks of $Clic5^{+/-}$ ($n = 12$), $Clic5^{+/-}$ injected with ssAAV.Clic5 ($n = 10$) or with scAAV.Clic5 ($n = 14$). (E) Distance traveled in the open-field test of $Clic5^{+/-}$ ($n = 16$) and $Clic5^{+/-}$ injected with ssAAV.Clic5 ($n = 14$) or scAAV.Clic5 ($n = 18$), performed at 4 weeks. (F) Quantification of circling behavior during the open field test of $Clic5^{+/-}$ ($n = 16$) and $Clic5^{+/-}$ injected with ssAAV.Clic5 ($n = 14$) or scAAV.Clic5 ($n = 18$), performed at 4 weeks. (G) Quantification of the average duration on the rotarod of $Clic5^{+/-}$ ($n = 8$) and $Clic5^{+/-}$ injected with ssAAV.Clic5 ($n = 8$) or scAAV.Clic5 ($n = 8$), performed at 8 weeks. Data information: The statistical tests were two-way ANOVA with Holm–Sidak correction for multiple comparisons for (A–D) and one-way ANOVA followed by Tukey correction for multiple comparisons for (E–G). Plots show mean ± SD. ns = not significant. Source data are available online for this figure.

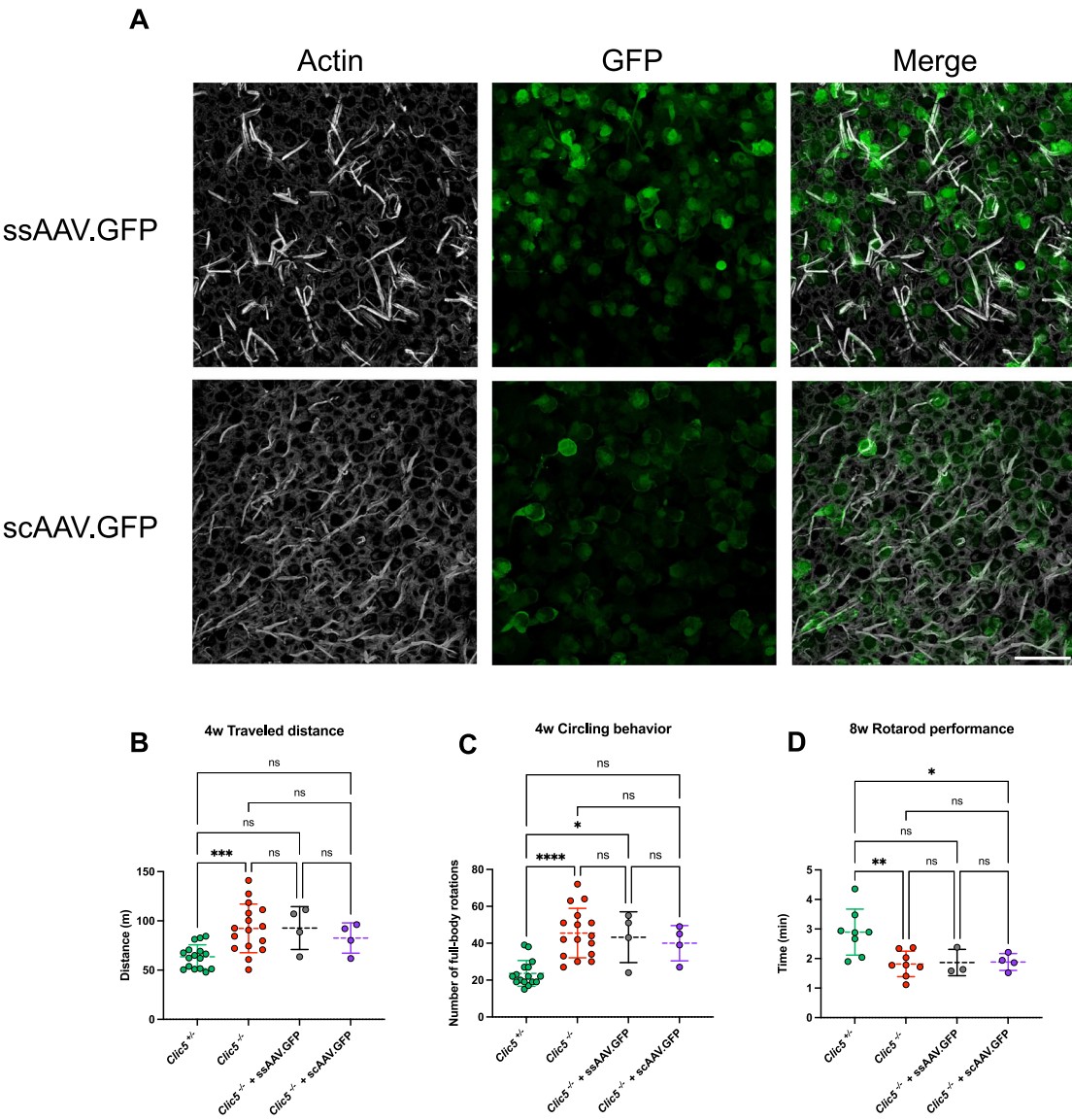

**Figure EV3.** **Single-stranded or self-complementary AAV.GFP does not rescue the morphology or function of the vestibular system.**

(A) High-magnification image of utricular hair cells of 12 weeks *Clic5⁻/⁻* injected with ssAAV.GFP or scAAV.GFP. (B) Distance traveled in the open-field test of *Clic5⁺/⁻* ($n = 16$), *Clic5⁻/⁻* ($n = 16$), *Clic5⁻/⁻* injected with ssAAV.GFP ($n = 4$), or scAAV.GFP ($n = 4$), performed at 4 weeks. (C) Quantification of circling behavior during the open field test of *Clic5⁺/⁻* ($n = 16$), *Clic5⁻/⁻* ($n = 16$), *Clic5⁻/⁻* injected with ssAAV.GFP ($n = 4$), or scAAV.GFP ($n = 4$), performed at 4 weeks. (D) Quantification of the average duration on the rotarod of *Clic5⁺/⁻* ($n = 8$), *Clic5⁻/⁻* ($n = 8$), *Clic5⁻/⁻* injected with ssAAV.GFP ($n = 3$), or scAAV.GFP ($n = 4$), performed at 8 weeks. Data information: The statistics test was one-way ANOVA followed by Tukey correction for multiple comparisons. Plots show mean ± SD. ns = not significant, *$P < 0.05$, **$P < 0.01$, ***$P < 0.001$, ****$P < 0.0001$ (Exact $P$ values are provided in Appendix Table S1). Scale bar = 10 µm. Source data are available online for this figure.

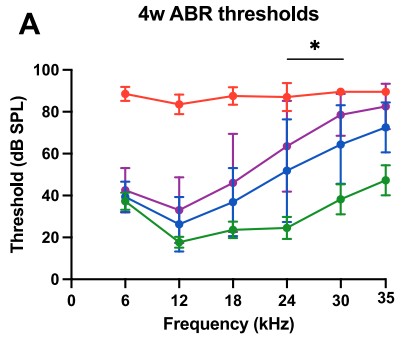

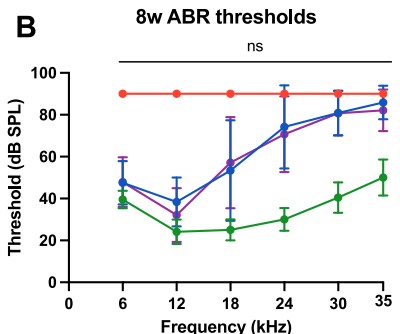

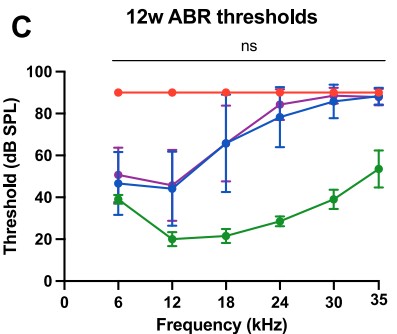

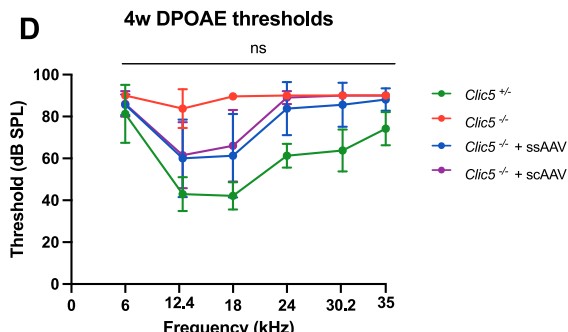

**Figure EV4.  Comparison of single-stranded and self-complementary vectors for auditory function restoration.**

(A) ABR thresholds at 4 weeks of *Clic5*+/− (*n* = 11), *Clic5*−/− (*n* = 10), and *Clic5*−/− injected with ssAAV.*Clic5* (*n* = 8) or scAAV.*Clic5* (*n* = 10). (B) ABR thresholds at 8 weeks of *Clic5*+/− (*n* = 11), *Clic5*−/− (*n* = 11), and *Clic5*−/− injected with ssAVV.*Clic5* (*n* = 6) or scAAV.*Clic5* (*n* = 7). (C) ABR thresholds at 12 weeks of *Clic5*+/− (*n* = 10), *Clic5*−/− (*n* = 10), and *Clic5*−/− injected with ssAAV.*Clic5* (*n* = 6) or scAAV.*Clic5* (*n* = 7). (D) DPOAE thresholds at 4 weeks of *Clic5*+/− (*n* = 12), *Clic5*−/− (*n* = 12), and *Clic5*−/− injected with ssAVV.*Clic5* (*n* = 8) or scAAV.*Clic5* (*n* = 10). Data information: Statistical tests were two-way ANOVA with Holm–Sidak correction for multiple comparisons. Plots show mean ± SD. ns = not significant, *P < 0.05 (Exact P values are provided in Appendix Table S1). Source data are available online for this figure.

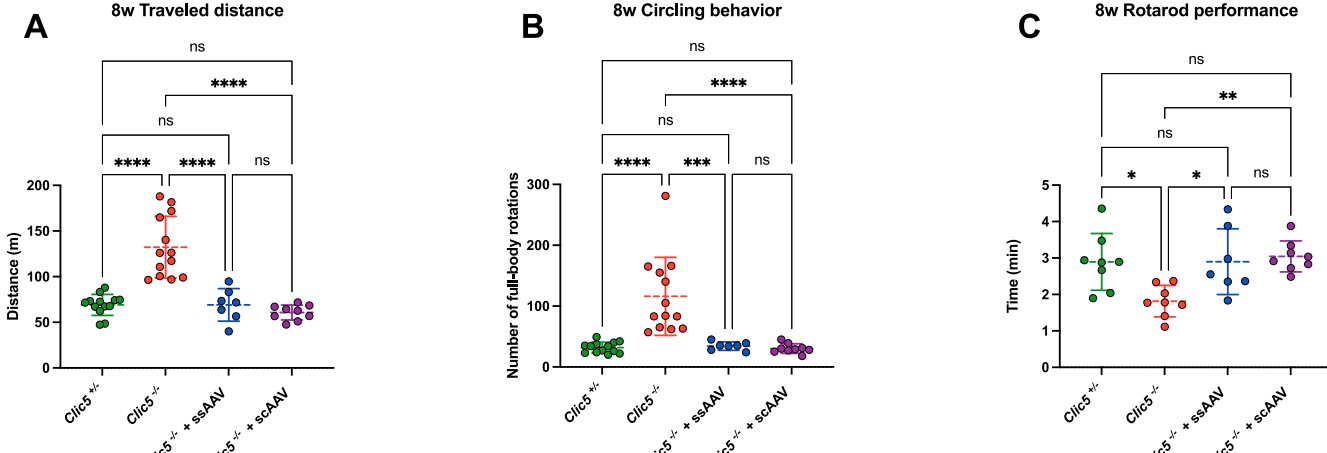

**Figure EV5. Comparison of single-stranded and self-complementary vectors for vestibular function restoration.**

(A) Distance traveled in the open-field test of $Clic5^{+/-}$ ($n = 13$), $Clic5^{-/-}$ ($n = 13$), and $Clic5^{-/-}$ injected with ssAAV.Clic5 ($n = 7$) or scAAV.Clic5 ($n = 9$), performed at 8 weeks. (B) Quantification of circling behavior during the open field test $Clic5^{+/-}$ ($n = 13$), $Clic5^{-/-}$ ($n = 13$), and $Clic5^{-/-}$ injected with ssAAV.Clic5 ($n = 7$) or scAAV.Clic5 ($n = 9$), performed at 8 weeks. (C) Quantification of the average duration on the rotarod of $Clic5^{+/-}$ ($n = 8$), $Clic5^{-/-}$ ($n = 8$), and $Clic5^{-/-}$ injected with ssAAV.Clic5 ($n = 7$) or scAAV.Clic5 ($n = 8$), performed at 8 weeks. Data information: The statistical test was One-way ANOVA followed by Tukey correction for multiple comparisons. Plots show mean ± SD. ns = not significant, $*P < 0.05$, $**P < 0.01$, $***P < 0.001$, $****P < 0.0001$ (Exact $P$ values are provided in Appendix Table S1). Source data are available online for this figure.

