## [Peer Review File · EMBO Molecular Medicine]

AAV gene therapy rescues hearing and balance in a model of CLIC5 deafness

Roni Hahn, Shahar Taiber, Olga Shubina-Oleinik, Gwenaelle Geleoc, Jeffrey Holt, and Karen Avraham

Corresponding author: Karen Avraham (karena@tauex.tau.ac.il)

Review Timeline:

Submission Date:	13th Feb 25
Editorial Decision:	11th Mar 25
Revision Received:	7th Jun 25
Editorial Decision:	2nd Jul 25
Revision Received:	8th Jul 25
Accepted:	9th Jul 25

Editor: Zeljko Durdevic

Transaction Report:

11th Mar 2025

Dear Prof. Avraham,

Thank you for the submission of your manuscript to EMBO Molecular Medicine. We have now received feedback from the three reviewers who agreed to evaluate your manuscript. All three referees recognize interest of the study but also raise important concerns that should be addressed in a major revision. If you would like to discuss further the points raised by the referees, I am available to do so via email or video. Let me know if you are interested in this option.

We would welcome the submission of a revised version within three months for further consideration. Please let us know if you require longer to complete the revision.

I look forward to receiving your revised manuscript.

Yours sincerely,

Zeljko Durdevic

We require:

- 1) A .docx formatted version of the manuscript text (including legends for main figures, EV figures and tables). Please make sure that the changes are highlighted to be clearly visible.
- 2) Individual production quality figure files as .eps, .tif, .jpg (one file per figure). For guidance, download the 'Figure Guide PDF': (<https://www.embopress.org/page/journal/17574684/authorguide#figureformat>).
- 3) A .docx formatted letter INCLUDING the reviewers' reports and your detailed point-by-point responses to their comments. As part of the EMBO Press transparent editorial process, the point-by-point response is part of the Review Process File (RPF), which will be published alongside your paper.
- 4) A complete author checklist, which you can download from our author guidelines (<https://www.embopress.org/page/journal/17574684/authorguide#submissionofrevisions>). Please insert information in the checklist that is also reflected in the manuscript. The completed author checklist will also be part of the RPF.
- 5) Please note that all corresponding authors are required to supply an ORCID ID for their name upon submission of a revised manuscript.
- 6) It is mandatory to include a 'Data Availability' section after the Materials and Methods. Before submitting your revision, primary

datasets produced in this study need to be deposited in an appropriate public database, and the accession numbers and database listed under 'Data Availability'. Please remember to provide a reviewer password if the datasets are not yet public (see <https://www.embopress.org/page/journal/17574684/authorguide#dataavailability>).

12) Author contributions: You will be asked to provide CRediT (Contributor Role Taxonomy) terms in the submission system. These replace a narrative author contribution section in the manuscript.

13) A Conflict of Interest statement should be provided in the main text.

14) Every published paper now includes a 'Synopsis' to further enhance discoverability. Synopses are displayed on the journal webpage and are freely accessible to all readers. They include a short stand first (maximum of 300 characters, including space) as well as 2-5 one-sentences bullet points that summarizes the paper. Please write the bullet points to summarize the key NEW findings. They should be designed to be complementary to the abstract - i.e. not repeat the same text. We encourage inclusion

of key acronyms and quantitative information (maximum of 30 words / bullet point). Please use the passive voice. Please attach these in a separate file or send them by email, we will incorporate them accordingly.

15) Include a Reagents and Tools Table as part of the Methods section, which can be downloaded from our author guidelines (<https://www.embopress.org/page/journal/17574684/authorguide#structuredmethods>)

**** Reviewer's comments ****

Referee #1 (Comments on Novelty/Model System for Author):

This is a nicely designed study. The technical quality of the results is high. AAV-mediated gene therapy has been applied to various mouse models of hereditary hearing loss, though this is the first report of its application in a mouse model of DFNB103. The testing of scAAV vs ssAAV is novel and interesting.

Referee #1 (Remarks for Author):

In this study, the authors tested AAV-mediated inner ear gene therapy in a mouse model of DFNB103, which is caused by mutations in the CLIC5 gene. When AAV2/9-PHP.B-CB6-Clc5 was injected into the utricle of P0 Clc5^{-/-} mice, they found restoration of CLIC5 expression at the stereocilia base in cochlear and vestibular hair cells. The stereocilia morphology and OHC survival were also improved. In addition, these mutant mice had improvement in auditory and vestibular functions. The authors also tested whether single-stranded construct vs. self-complementary construct had any effect on the treatment outcomes, and found that both of these constructs were effective at improving the auditory and vestibular functions in these mutant mice, though the self-complementary construct achieved comparable functional recovery at a lower viral titer compared to the single-stranded construct. The study was well-designed and the experiments were technically sound. My comments are listed below:

1. The CB6 promoter was used to drive CLIC5 expression. Was ectopic expression of CLIC5 found in other cell types in the inner ear? Or was CLIC5 expression only seen in hair cells?
2. In Figure 2A, is there some residual expression of CLIC5 in the mutant?
3. In Figure 3A, what does the green channel represent? How was hair cell survival quantified?
4. In Figure 4F, the freezing duration for the "before" non-treated mutants seems to be much higher than the "before" scAAV-treated mice and also the "before" untreated heterozygous littermates. Were the differences in the "before" freezing duration between these three animal groups statistically significant?
5. The authors pointed out that the Clc5^{-/-} mice show vestibular hair cell degeneration, as evidenced by stereocilia elongation. Was the actual number of vestibular hair cells decreased as well? If so, was the survival of vestibular hair cells improved with gene therapy?
6. In Figure EV1B, there seems to be an improvement in 4w DPOAE threshold at 12.4 kHz in the scAAV-GFP group. What would be the reason for this?
7. As the authors pointed out in the introduction, CLIC5 has been shown to interact with other hair cell proteins, such as RDX, PTPRQ, and TPRN. Does the gene therapy treatment also restore the organization of these proteins at the stereocilia base?
8. A recent study by Qi et al. (Mol Ther 2024) showed that proper amount of TPRN expression is important for hearing recovery in a Tprn KO mouse model. Was this dose dependence also observed with CLIC5 expression?

Referee #2 (Comments on Novelty/Model System for Author):

Appropriate work using appropriate model, and cutting edge technological approaches and tools

Referee #2 (Remarks for Author):

In this manuscript, Roni Hahn et al. assess the therapeutic potential of adeno-associated virus (AAV)-based gene therapy for DFNB103-related hearing and vestibular dysfunction, caused by mutations in the Clc5 gene. The study focuses on comparing single-stranded AAV (ssAAV) and self-complementary AAV (scAAV) for delivering the wild-type Clc5 gene in a Clc5-deficient mouse model carrying the c.680T>C pathogenic variant (p.(Phe318Ser); Jax #005329). The scAAV has been reported to enable

a faster and more robust onset of transgene expression relative to traditional single-stranded (ss) AAV vectors.

The key findings of the study include:

- Successful restoration of *Clic5* expression in inner ear hair cells using both ssAAV and scAAV vectors with the AAV9-PHP capsid.
- Prevention of hair cell degeneration, with improved morphology and survival of auditory hair cells.
- Significant recovery of auditory function (via ABR and DPOAE measures) in treated mice, particularly at low-to-mid frequencies and at early stages post-injection.
- Restoration of vestibular hair cell structure and associated balance function, reducing circling behavior and hyperactivity in treated mice.

The comparative analysis of ssAAV and scAAV vectors, including the demonstration that scAAV achieves similar functional rescue at a significantly lower dose, is particularly valuable for translational applications.

Overall, this manuscript presents significant data demonstrating that *Clic5* gene replacement via ssAAV and scAAV AAV9-PHP vectors can restore hearing and balance functions in *Clic5*-deficient mice. These findings meaningfully contribute to the field of inner ear gene therapy, supporting the feasibility of *Clic5* gene replacement as a potential therapeutic approach. Some aspects of the study however require further clarification and additional data presentation to strengthen the conclusions.

Addressing the following points would further improve the manuscript:

1. While balance function assays are well documented, data on GFP transduction rates in vestibular organs and differences in expression between striolar and extrastriolar regions are missing. Including these data would provide a more precise understanding of vector distribution and efficacy, helping to correlate with representative SEM images in the figures.
2. The study presents both ABR and DPOAE data, but DPOAEs are only shown up to 4 weeks post-injection. DPOAE measurements at 8 and 12 weeks or DPOAE amplitudes for a given frequency at 4, 8, and 12 weeks would provide a clearer picture of cochlear function over time, similar to the balance data in Fig. 5K and L.
3. The study evaluates auditory fear conditioning in twelve-week-old mice but only at 6 kHz. Testing and presenting data with mid- and high-frequency tones would help correlate behavioral responses with ABR and DPOAE results.
4. The sequence of pathogenic changes in *Clic5*-deficient mice before 4 weeks should be clarified to better contextualize the therapeutic window and differences in recovery between frequencies.
5. The study demonstrates that scAAV is effective at an order of magnitude lower titer than ssAAV (1.52×10^{13} gc/ml vs. 1.69×10^{14} gc/ml). However, since 1.52×10^{13} gc/ml is still a relatively high dose for AAV therapy, testing lower doses would help determine the minimum effective dose and its impact on immunogenicity. Have the authors confirmed the non-toxicity of the therapeutic vector upon injection in wild-type mice?

Minor Comments:

For all figures, the stage of analysis (e.g., 4 weeks, 8 weeks, or 12 weeks) should be clearly indicated. Use "Fig XA, and B" instead of "Figs (plural)".

Figure 2: The current low-magnification IF images do not clearly illustrate differences between genotypes and treatment conditions. Higher magnification images of F-actin-stained organ of Corti, or other hair bundle biomarkers, with zoom-ins on OHCs and IHCs, would enhance clarity.

Figure 3: If quantification is based on SEM images, it should refer to hair bundle counts rather than hair cells (which should be counted using MYO7A-labeled cells). The SEM panel of OHC bundles appears to show a mid-cochlear region; indicating the specific frequency region (e.g., 24 or 32 kHz) instead of labeling it as the cochlear base (50 or 60 kHz) would be more precise.

Figure 4: Including a longitudinal plot for a representative frequency (e.g., 18 kHz) across different time points (4, 8, and 12 weeks) would better illustrate progressive changes. Missing DPOAE data at 8 and 12 weeks could be included.

Figure 5: Higher magnification images of utricular VHCs should be provided, with zoomed-in staining of a VHC in a specific utricular region (indicating position: striola or extrastriola). The middle panel in Fig. 5C is unclear; a low-magnification SEM image could improve clarity.

Figure 6: The GFP expression pattern in panel A appears to extend beyond the organ of Corti region-clarifying this observation is necessary, as GFP staining in auditory hair cells appears lower.

Figure 7: There seems to be a labeling error in one of the panels, which should depict untreated *Clic5*^{-/-} mice. Furthermore, the IHC and OHC bundle images in panel G display pathogenic features inconsistent with normal hearing heterozygous *Clic5*^{+/-} mice.

Figure 8: To improve clarity, combining Figures 5 and 8 into one or two figures that directly compare ssAAV and scAAV treatment outcomes would be more informative.

Referee #3 (Remarks for Author):

In this preclinical study Hahn et al. investigated the potential for AAV gene therapy for restoring hearing and balance in a mouse model of human DFNB103 induced by mutations in *CLIC5*. *Clic5*-deficient mice show deafness and circling behavior. The study used AAV to introduce the wild-type *Clic5* coding sequence into *Clic5*-deficient mice by single-stranded and self-complementary AAVs. Both approaches led to robust transgenic *Clic5* expression which restored auditory and vestibular function and prevented

morphological

degeneration and preserving. The self-complementary AAVs achieved comparable functional recovery at a lower titer than with the single-stranded construct. This work contributes to our understanding of inherited inner ear disease and shows promise for self-complementary AAVs in inner ear gene therapy.

This is a well-executed study and the MS is overall well-prepared. I have mostly minor suggestions for how to further improve the MS.

I have four general comments:

- Please consider careful wording when discussing preclinical work on rodents in the framework of inner ear gene therapy. P0 injections of viral vectors into the inner ear of mice are an important method, but I recommend not phrasing this "gene therapy" to avoid misconception.
- I recommend more balanced referencing. For example, I was surprised for a preclinical study to miss citing the pioneering preclinical work on OTOF gene therapy (Akil et al., 2019; Al-Moyed et al., 2019) on which the clinical trials rest. This would probably be important (perhaps more than quoting the reviews). Moreover, the original publication of AAV9-PHP.B (Deverman 2016) should be cited. Also, from a clinical point of view the route of delivering AAV to the inner ear via the cerebrospinal fluid (top of page 4) seems obsolete.
- I suggest to use a better allele name than *Clic5*^{-/-} (which implies knockout of the gene) for this point mutant.
- The poorer functional outcome at later points after early postnatal injection should be discussed: could there be lower transgene expression at these later times?

Specific comments:

- Please make sure to consistently use gene and protein nomenclature (mouse/human) as appropriate for the context
- I suggest to add the age of mice analyzed for Figs. 2 and 3.
- It did not become entirely clear to me whether the authors differentiate between "restore" and "rescue". Both are used throughout the MS and differentially for figure titles.
- This seems to be a fragment: "This outcome has also been observed in the inner ear with AAV vectors driving green fluorescent protein (GFP) expression in cochlear explants and in vivo with cochlear injections (Casey et al, 2020; Maguire & Corey, 2020),..." I also wonder whether the Maguire & Corey review is appropriate as reference here.
- "DPOAE results also indicated significant differences": specify differences
- "Gene therapy restores CLIC5 expression in the auditory system" see general comment
- "significantly elongated hair bundles compared to control heterozygous mice, especially in IHCs (Figs 3A and B; $p < 0.0001$ " What measurement does the p value relate to?
- "Quantification of HC survival indicated that CLIC5 deficiency primarily caused OHC loss" see general comment
- "To evaluate the therapeutic effect of gene replacement therapy on auditory function, ABRs and DPOAEs were recorded at different time points in mice that received injections of ssAAV.Clic5" specify age of injection
- Among the GFP-positive OHCs, the mean fluorescence intensity of scAAV was approximately three times greater than that of ssAAV (28.14 for ssAAV vs. 83.60 for scAAV) (Fig 6D). add arbitrary units!

Dear Dr. Durdevic and the Editorial Board,

Thank you for evaluating our manuscript entitled "AAV gene therapy rescues hearing and balance in a model of *CLIC5* deafness" for publication in *EMBO Molecular Medicine*.

Please find below our detailed responses to the reviewers. The recommended changes are listed below and marked in red font in the text file.

We very much hope that these responses will allow our manuscript to be published in *EMBO Molecular Medicine*.

Sincerely,

Karen

Prof. Karen Avraham, Ph.D.

Drs. Sarah and Felix Dumont Chair for Research of Hearing Disorders

Dean, Gray Faculty of Medical and Health Sciences

Laboratory for Neural and Sensory Genomics

Department of Human Molecular Genetics and Biochemistry

Gray Faculty of Medical and Health Sciences and Sagol School of Neuroscience

Tel Aviv University

Referee #1 (Comments on Novelty/Model System for Author):

This is a nicely designed study. The technical quality of the results is high. AAV-mediated gene therapy has been applied to various mouse models of hereditary hearing loss, though this is the first report of its application in a mouse model of DFN103. The testing of scAAV vs ssAAV is novel and interesting.

Referee #1 (Remarks for Author):

In this study, the authors tested AAV-mediated inner ear gene therapy in a mouse model of DFN103, which is caused by mutations in the *CLIC5* gene. When AAV2/9-PHP.B-CB6-*Clic5* was injected into the utricle of P0 *Clic5*^{-/-} mice, they found restoration of *CLIC5* expression at the stereocilia base in cochlear and vestibular hair cells. The stereocilia morphology and OHC survival were also improved. In addition, these mutant mice had improvement in auditory and vestibular functions. The authors also tested whether single-stranded construct vs. self-complementary construct had any effect on the treatment outcomes, and found that both of these constructs were effective at improving the auditory and vestibular functions in these mutant mice, though the self-complementary construct achieved comparable functional recovery at a lower viral titer compared to the single-stranded construct. The study was well-designed and the experiments were technically sound. My comments are listed below:

1. The CB6 promoter was used to drive CLIC5 expression. Was ectopic expression of CLIC5 found in other cell types in the inner ear? Or was CLIC5 expression only seen in hair cells?

Response: Based on previous studies (Gagnon et al, 2006), our analysis, and gEAR data, CLIC5 expression is specific to hair cells in the inner ear, with no expression reported in other cell types within the inner ear. Similarly, following gene therapy, *Clic5* expression remained specific to hair cells.

2. In Figure 2A, is there some residual expression of CLIC5 in the mutant?

Response: The red dots in the image are likely background noise and do not indicate a significant signal. This was confirmed by comparing the staining pattern to negative controls and quantifying the differences in signal intensity between control and mutant samples. In addition, the antibody that was used targets a sequence located downstream of the mutation site.

3. In Figure 3A, what does the green channel represent? How was hair cell survival quantified?

Response: The green channel represents actin staining (Phalloidin). Hair cell survival was quantified by Myosin VIIa staining. Images were taken from different regions of the cochlea and counted manually. Each sample included three replicates (3 different mice).

The following text has been added to the Methods section:

Hair cell survival was assessed using Myosin VIIa staining, and counts were performed on the same-sized segments (92.26 μm) from different cochlear regions.

4. In Figure 4F, the freezing duration for the "before" non-treated mutants seems to be much higher than the "before" scAAV-treated mice and also the "before" untreated heterozygous littermates. Were the differences in the "before" freezing duration between these three animal groups statistically significant?

Response: We thank the reviewer for this comment. As noted, the mice exhibit freezing behavior even without a tone. This suggests that memory formation remains intact despite the loss of *Clic5*. However, only the control mice (heterozygous) and the treated *Clic5*^{-/-} mice freeze significantly more after the tone than before. This indicates that they are able to hear the tone and associate it with the electric shock. On the other hand, while *Clic5*^{-/-} cannot hear the tone and therefore respond to the setting (i.e., the cage, the smells, etc.), and exhibit freezing behavior that is not associated with the appearance of the tone. The activity level is measured automatically. It shows the opposite of freezing: when mice hear the tone, their activity decreases significantly. Similar results are found in a previous study performed by our group (Taiber et al, 2020).

5. The authors pointed out that the *Clic5*^{-/-} mice show vestibular hair cell degeneration, as evidenced by stereocilia elongation. Was the actual number of vestibular hair cells decreased as well? If so, was the survival of vestibular hair cells improved with gene therapy?

Response: We thank the reviewer for this comment. We have now analyzed distinct regions of the utricle and quantified vestibular hair cells based on Myosin VIIa staining. The number of vestibular hair cells is reduced in mutant mice compared to controls. Following treatment with each AAV, hair cell numbers are restored to levels comparable to those of the control group.

The following text has been added to the Results section:

...Additionally, the number of VHCs, which were reduced in mutant mice, was restored to near-control levels following treatment with ssAAV vector (Fig 5C).

.... and improved VHC survival, similar to the results observed with ssAAV treatment (Fig 8A-C).

In the Methods:

To quantify the number of VHC, hair cells were labeled using Myosin VIIa staining. Two images were acquired from consistent regions of each utricle on the same-sized segments (92.26 μm), and Myosin VIIa-positive hair cells were counted in each image. The average of the two counts was used as a single biological replicate for each sample.

In the figure legend:

Figure 5:

C Quantification of VHC survival in mouse utricles at 12 weeks.

Figure 8:

C Quantification of VHC survival in mouse utricles at 12 weeks.

6. In Figure EV1B, there seems to be an improvement in 4w DPOAE threshold at 12.4 kHz in the scAAV-GFP group. What would be the reason for this?

Response: The scAAV-GFP group included 4 individuals, two of which had high threshold levels (12.4 kHz = 90, 75), while the other two had lower thresholds (12.4 kHz = 45, 65). Since no such variability was observed at 8 or 12 weeks of age during ABR testing, which showed complete hearing loss across all frequencies, we believe this observation reflects variability in the phenotype of the mice.

7. As the authors pointed out in the introduction, CLIC5 has been shown to interact with other hair cell proteins, such as RDX, PTPRQ, and TPRN. Does the gene therapy treatment also restore the organization of these proteins at the stereocilia base ?

Response: We thank the reviewer for this excellent comment. We have now analyzed the localization of the hair cell proteins RDX and TPRN in mice treated with ssAAV.*Clic5* (1.69×10^{13} gc/ml) and tested 8 weeks post-injection. In these mice, RDX expression remained primarily restricted to the base of the stereocilia, similar to the control group. In contrast, TPRN expression was distributed throughout the stereocilia. This may result from the effect of *Clic5* overexpression. Similar results were reported by Qi et al. (Mol Ther 2024), who showed that overexpression of TPRN led to its mislocalization, extending from the base to the tip of the stereocilia.

The following text has been added to the Results section:

RDX and TPRN, similar to CLIC5, are proteins expressed in HCs and are normally localized at the base of the stereocilia. Previous studies have shown that the absence of CLIC5 disrupts this localization, resulting in the distribution of RDX and TPRN along the stereocilia, which may contribute to the progressive degeneration of the hair bundle structure in *Clic5*^{-/-} mice (Salles *et al.*, 2014) (Fig 2D and E). To test whether restoring CLIC5 expression could restore the proper localization of these proteins, *Clic5*^{-/-} mice were injected with ssAAV.*Clic5* (titer 1.69×10^{13} gc/ml) at P0 and harvested 8 weeks post injection. RDX expression was concentrated at the base of the hair bundle in treated mice, similar to the pattern observed in *Clic5*^{+/+} control (Fig 2D). In contrast, TPRN was mislocalized along the entire length of the stereocilium, rather than being concentrated at the base (Fig 2E). A similar mislocalization pattern was reported by Qi et al. (Mol Ther, 2024), where AAV-mediated TPRN overexpression disrupted its typical localization.

In the Methods:

... rabbit anti-TPRN (Sigma HPA020899) 1:200, rabbit anti-RDX (Sigma R3653) 1:200.

In the figure legend:

D Representative confocal images of IHCs from the apical turn of *Clic5*^{+/+}, *Clic5*^{-/-}, and *Clic5*^{-/-} injected with scAAV.*Clic5* at 8 weeks, stained with a RDX (magenta) and actin (cyan).

E Representative confocal images of IHCs from the apical turn of *Clic5*^{+/+}, *Clic5*^{-/-}, and *Clic5*^{-/-} injected with scAAV.*Clic5* at 8 weeks, stained with a TPRN (magenta) and actin (cyan).

In the Discussion:

... Restoring CLIC5 expression also appears to normalize the localization of RDX, emphasizing CLIC5's vital role in hair cell function.

8. A recent study by Qi et al. (Mol Ther 2024) showed that proper amount of TPRN expression is important for hearing recovery in a Tprn KO mouse model. Was this dose dependence also observed with CLIC5 expression?

Response: We thank the reviewer for this important comment. We have now analyzed three different titers of scAAV.*Clic5*: The original titer (1.52×10^{13} gc/ml) and two other diluted concentrations (10^{12} and 10^{11} gc/ml). Our results indicated a dose-dependent effect, with the original higher titer resulting in a greater number of Flag-positive hair cells and better hearing rescue. The two lower doses mediated partial rescue, primarily at low frequencies.

The following text has been added to the Results section:

To evaluate the minimum effective dose of scAAV.*Clic5*, inner ear injections were performed, as previously described, in *Clic5*^{-/-} mice using three different viral titers: the original concentration (1.52×10^{13} gc/ml) and two lower doses (1.52×10^{12} and 1.52×10^{11} gc/ml). Four weeks post-injection, auditory function was assessed, and the inner ear was collected for immunostaining. Due to the size limitations of scAAV, a Flag tag was used to track expression instead of GFP. Expression efficiency was quantified by calculating the percentage of Flag-positive cells among Myosin VIIa-labeled hair cells. The number of positive hair cells in treated mice increased as a function of the titer (Fig 7O). When tested for ABR, mice that received the highest titer (1.52×10^{13} gc/ml) exhibited improved hearing sensitivity across a broader range of frequencies (Fig 7P). The two lower doses resulted in similar thresholds, with rescue primarily limited to low frequencies. Taken together, rescue by scAAV.*Clic5* seems to be dose-dependent.

In the figure legend:

Figure 7:

O Expression rates of scAAV.*Clic5* across the cochlea as a function of viral titer, n=3 for each group

P ABR thresholds at 4 weeks of *Clic5*^{-/-} injected with scAAV.*Clic5* at different viral doses: 1.52×10^{13} gc/ml (n=10), 1.52×10^{12} gc/ml (n=3) and 1.52×10^{11} gc/ml (n=3).

In the Discussion:

Our dose-response analysis showed that even reduced titers of scAAV.*Clic5* can mediate partial hearing restoration (at levels as low as 1.52×10^{11} gc/ml), supporting the feasibility of lower dosing strategies.

Referee #2 (Comments on Novelty/Model System for Author):

Appropriate work using appropriate model, and cutting edge technological approaches and tools

Referee #2 (Remarks for Author):

In this manuscript, Roni Hahn et al. assess the therapeutic potential of adeno-associated virus (AAV)-based gene therapy for DFNB103-related hearing and vestibular dysfunction, caused by mutations in the *Clic5* gene. The study focuses on comparing single-stranded AAV (ssAAV) and self-complementary AAV (scAAV) for delivering the wild-type *Clic5* gene in a *Clic5*-deficient mouse model carrying the c.680T>C pathogenic variant (p.(Phe318Ser); Jax #005329). The scAAV has been reported to enable a faster and more robust onset of transgene expression relative to traditional single-stranded (ss) AAV vectors.

The key findings of the study include:

- Successful restoration of *Clic5* expression in inner ear hair cells using both ssAAV and scAAV vectors with the AAV9-PHP capsid.
- Prevention of hair cell degeneration, with improved morphology and survival of auditory hair cells.
- Significant recovery of auditory function (via ABR and DPOAE measures) in treated mice, particularly at low-to-mid frequencies and at early stages post-injection.
- Restoration of vestibular hair cell structure and associated balance function, reducing circling behavior and hyperactivity in treated mice.

The comparative analysis of ssAAV and scAAV vectors, including the demonstration that scAAV achieves similar functional rescue at a significantly lower dose, is particularly valuable for translational applications.

Overall, this manuscript presents significant data demonstrating that *Clic5* gene replacement via ssAAV and scAAV AAV9-PHP vectors can restore hearing and balance functions in *Clic5*-deficient mice. These findings meaningfully contribute to the field of inner ear gene therapy, supporting the feasibility of *Clic5* gene replacement as a potential therapeutic approach. Some aspects of the study however require further clarification and additional data presentation to strengthen the conclusions.

Addressing the following points would further improve the manuscript:

1. While balance function assays are well documented, data on GFP transduction rates in vestibular organs and differences in expression between striolar and extrastriolar regions are missing. Including these data would provide a more precise understanding of vector distribution and efficacy, helping to correlate with representative SEM images in the figures.

Response: We thank the reviewer for this constructive comment. We have now quantified GFP-positive VHCs in each sample by analyzing two images per utricle acquired from consistent areas. This quantification provides an estimate of overall expression efficiency in the utricle. However, we did not specifically distinguish between striolar and extrastriolar regions in this analysis. While we agree that such regional resolution could offer valuable insights into vector distribution and expression patterns, our study primarily focused on overall expression efficiency and associated functional outcomes.

The following text has been added to the Results section:

...Both AAVs demonstrated high expression efficiency in utricular VHC, with GFP positive cells in over 97% of VHCs ($p=0.4453$) (Fig 6A and B).

In the Methods:

...Similarly, GFP-positive VHCs were quantified using two images per utricle, acquired from consistent regions (92.26 μm segments), and the percentage was calculated relative to the total number of VHCs.

In the figure legend:

Figure 6:

A High-magnification images of VHCs from the utricle of *Clic5*^{+/+} injected with ssAAV.GFP or scAAV.GFP at p14.

B Expression rates of ssAAV.GFP and scAAV.GFP across the utricle, n=3 for each group.

2. The study presents both ABR and DPOAE data, but DPOAEs are only shown up to 4 weeks post-injection. DPOAE measurements at 8 and 12 weeks or DPOAE amplitudes for a given frequency at 4, 8, and 12 weeks would provide a clearer picture of cochlear function over time, similar to the balance data in Fig. 5K and L.

Response: We thank the reviewer for this comment. We found that DPOAE thresholds recovered to a similar extent as the ABR thresholds at 4 weeks. Since normal DPOAEs (i.e. outer hair cell function) are required for normal ABRs and since ABR thresholds remained relatively stable out to 12 weeks, the data implied that DPOAEs must have remained stable as well. As such, we opted to focus on other analyses, such as fear conditioning and morphological characterization. Had the ABRs been elevated at later stages, additional DPOAE analysis would have been warranted.

3. The study evaluates auditory fear conditioning in twelve-week-old mice but only at 6 kHz. Testing and presenting data with mid- and high-frequency tones would help correlate behavioral responses with ABR and DPOAE results.

Response: We thank the reviewer for this comment. Fear conditioning is a classical assay in which freezing behavior has been shown to depend on components of the auditory system in the CNS. Therefore, we are interested in examining whether the AAV.*Clic5* can affect auditory behaviors that require central auditory processing. In addition, from an ethical perspective, fear conditioning involved stimuli like shocks, and therefore, we were required by our Institutional Ethics Committee to perform fear conditioning as a terminal experiment, which prevented us from testing additional frequencies. We note that the fact that fear conditioning behavior was rescued answers a yes/no question regarding central processing of auditory cues. ABR and DPOAE were used to assess cochlear function in more detail.

4. The sequence of pathogenic changes in *Clic5*-deficient mice before 4 weeks should be clarified to better contextualize the therapeutic window and differences in recovery between frequencies.

Response: We thank the reviewer for this important comment. Additional analysis was performed using scanning electron microscopy. Changes in the morphology of *Clic5*^{-/-} became more pronounced by P17, at which IHCs exhibited abnormal and fused hair bundles across all three regions. OHCs were primarily affected in the apical region, but abnormalities were also observed in the mid and basal regions. For VHCs, morphological abnormalities of the hair bundles were present as early as p15.

The following text has been added to the Results section:

Morphological changes were studied using scanning electron microscopy (SEM). At postnatal day 17 (P17), hair cell abnormalities were evident in the auditory system, with outer hair cell (OHC) bundles appearing fused and disorganized, particularly in the apical region (Fig 1H and I). Inner hair cells (IHCs) exhibited extensive hair bundle fusion across all cochlear regions. No morphological changes were observed at P14, suggesting that the onset of hair cell defects occurs between P14 and P17 (Fig 1I). In the vestibular system, vestibular hair cells (VHCs) of the utricle displayed elongated hair bundles as early as P15, and by P17, thickened hair bundles were also observed (Fig 1J).

In the figure legend:

Figure title: Functional and morphological characterization of *Clic5* mice

H Representative scanning electron microscopy images of IHCs and OHCs at P17 at different regions of the cochlea of *Clic5*^{-/-}.

I High-magnification images of outer and inner hair bundles from the cochlear apex of *Clic5*^{-/-} at P14 and P17.

J High-magnification images of VHCs from the utricle of *Clic5*^{-/-} at P15 and P17.

In the Discussion:

...In this study, we performed utricle injections into P0 mice, prior to the morphological changes in hair cells that begin between P14 and P17, at the time of hearing onset.

5. The study demonstrates that scAAV is effective at an order of magnitude lower titer than ssAAV (1.52×10^{13} gc/ml vs. 1.69×10^{14} gc/ml). However, since 1.52×10^{13} gc/ml is still a relatively high dose for AAV therapy, testing lower doses would help determine the minimum effective dose and its impact on immunogenicity. Have the authors confirmed the non-toxicity of the therapeutic vector upon injection in wild-type mice?

Response: We thank the reviewer for this excellent comment. The heterozygous mice group served as the control in all experiments, as no differences in ABR and DPOAE threshold measurements were observed between wild-type and heterozygous mice. For the heterozygous group, safety and toxicity assessments were performed for both vectors as shown in figure EV2, with no effects on hearing or vestibular function in the control group.

The suggestion to test lower doses of scAAV has been addressed in response to a similar comment from another reviewer, and appropriate data were added (last comment of Referee #1).

Minor Comments:

For all figures, the stage of analysis (e.g., 4 weeks, 8 weeks, or 12 weeks) should be clearly indicated. Use "Fig XA, and B" instead of "Figs (plural)".

Response: We have included specific ages in the figure legends of Fig. 3B–D and Fig. 6B, and the figure citations have been updated accordingly.

Figure 2: The current low-magnification IF images do not clearly illustrate differences between genotypes and treatment conditions. Higher magnification images of F-actin-stained organ of Corti, or other hair bundle biomarkers, with zoom-ins on OHCs and IHCs, would enhance clarity.

Response: We thank the reviewer for this comment. We have now added an analysis that includes the organization of the hair cell proteins RDX and TPRN (linked to *Clic5*), which also highlights the

differences between the various genotypes more clearly. These results are discussed in response to Comment 7 from Referee #1.

Figure 3: If quantification is based on SEM images, it should refer to hair bundle counts rather than hair cells (which should be counted using MYO7A-labeled cells). The SEM panel of OHC bundles appears to show a mid-cochlear region; indicating the specific frequency region (e.g., 24 or 32 kHz) instead of labeling it as the cochlear base (50 or 60 kHz) would be more precise.

Response: Hair cell counts were performed based on the immunofluorescence images using Myo7a staining. In addition, the specific frequency of the SEM image was added to the legend of figure 3.

The following text has been added to the Methods section:

Hair cell survival was assessed using Myosin VIIa staining, and counts were performed on the same-sized segments (92.26 μm) from different cochlear regions.

In the figure legend:

Figure 3:

F High-magnification images of outer and inner hair bundles from the cochlear 32 kHz region of the *Clic5*^{-/-} and *Clic5*^{-/-} injected with ssAAV.*Clic5* at 12 weeks.

Figure 4: Including a longitudinal plot for a representative frequency (e.g., 18 kHz) across different time points (4, 8, and 12 weeks) would better illustrate progressive changes. Missing DPOAE data at 8 and 12 weeks could be included.

Response: We thank the reviewer for this comment. An analysis including the ABR thresholds of all frequencies at three different time points has been added.

The following text has been added to the Results section:

... Over time, the hearing sensitivity of the mutant mice treated with ssAAV.*Clic5* mainly remained stable at low frequencies (Fig 4F).

... Hearing sensitivity remained consistent at 4 and 8 weeks but decreased slightly by 12 weeks (Fig 7M).

In the figure legend:

Figure 4:

F ABR thresholds at 3-time points of *Clic5*^{-/-} injected with ssAAV.*Clic5*: 4 weeks (n=8), 8 weeks (n=6), and 12 weeks (n=6).

Figure 7:

M ABR thresholds at 3-time points of *Clic5*^{-/-} injected with scAAV.*Clic5*: 4 weeks (n=10), 8 weeks (n=7), and 12 weeks (n=7).

Figure 5: Higher magnification images of utricular VHCs should be provided, with zoomed-in staining of a VHC in a specific utricular region (indicating position: striola or extrastriola). The middle panel in Fig. 5C is unclear; a low-magnification SEM image could improve clarity.

Response: We thank the reviewer for this comment. The middle panel in Fig. 5C represents a single giant hair bundle. To clarify this point, we have added an illustration highlighting the differences in hair bundle size observed in the SEM images.

The following text has been added to the Results section:

... the morphology of utricular hair cells in untreated *Clic5*^{-/-} displayed fused giant hair bundles and lacked a regular staircase pattern (Fig 5D and E).

In the figure legend:

Figure 5:

E Illustration of hair bundle size differences observed in SEM Images

Figure 6: The GFP expression pattern in panel A appears to extend beyond the organ of Corti region-clarifying this observation is necessary, as GFP staining in auditory hair cells appears lower.

Response: The measurements of GFP expression (6E-6G) were based on whole-mount images, such as those shown in 6C. Since CLIC5 is mainly expressed in hair cells, the purpose of using the GFP construct was to assess whether the scAAV could efficiently and rapidly transfect hair cells, as shown previously in studies. Thus, the focus was not on characterizing potential scAAV.GFP expression in other cell types or regions.

Figure 7: There seems to be a labeling error in one of the panels, which should depict untreated *Clic5*^{-/-} mice. Furthermore, the IHC and OHC bundle images in panel G display pathogenic features inconsistent with normal hearing heterozygous *Clic5*^{+/-} mice.

Response: We thank the reviewer for pointing this out. The title of the panel G has been changed to *Clic5*^{-/-}.

Figure 8: To improve clarity, combining Figures 5 and 8 into one or two figures that directly compare ssAAV and scAAV treatment outcomes would be more informative.

Response: A similar comparison of the two treatment groups is presented in the extended view (Fig. EV5), which presents the vestibular function outcomes.

Referee #3 (Remarks for Author):

In this preclinical study Hahn et al. investigated the potential for AAV gene therapy for restoring hearing and balance in a mouse model of human DFNB103 induced by mutations in CLIC5. *Clic5*-deficient mice show deafness and circling behavior. The study used AAV to introduce the wild-type *Clic5* coding sequence into *Clic5*-deficient mice by single-stranded and self-complementary AAVs. Both approaches led to robust transgenic *Clic5* expression which restored auditory and vestibular function and prevented morphological degeneration and preserving. The self-complementary AAVs achieved comparable functional recovery at a lower titer than with the single-stranded construct. This work contributes to our understanding of inherited inner ear disease and shows promise for self-complementary AAVs in inner ear gene therapy. This is a well-executed study and the MS is overall well-prepared. I have mostly minor suggestions for how to further improve the MS. I have four general comments:

- Please consider careful wording when discussing preclinical work on rodents in the framework of inner ear gene therapy. PO injections of viral vectors into the inner ear of mice are an important method, but I recommend not phrasing this "gene therapy" to avoid misconception.

Response: Several papers have been published in *EMBO Molecular Medicine* covering injections of viral vectors into the inner ears of mice, using the term "gene therapy". These include: "Improved gene therapy for spinal muscular atrophy in mice using codon-optimized hSMN1 transgene and hSMN1 gene-derived promoter", *EMBO Mol Med* (2024) 16:945-965 and "Systemic gene therapy

rescues retinal dysfunction and hearing loss in a model of Norrie disease”, *EMBO Mol Med* (2023) 15:e17393

- I recommend more balanced referencing. For example, I was surprised for a preclinical study to miss citing the pioneering preclinical work on OTOF gene therapy (Akil et al., 2019; Al-Moyed et al., 2019) on which the clinical trials rest. This would probably be important (perhaps more than quoting the reviews). Moreover, the original publication of AAV9-PHP.B (Deverman 2016) should be cited. Also, from a clinical point of view the route of delivering AAV to the inner ear via the cerebrospinal fluid (top of page 4) seems obsolete.

Response: We have added the recommended references to the introduction (Akil et al., 2019; Al-Moyed et al., 2019; Deverman et al., 2016) and removed the example of cerebrospinal fluid.

- I suggest to use a better allele name than *Clic5*^{-/-} (which implies knockout of the gene) for this point mutant.

Response: The full approved name is C57BL/6J-*Clic5*^{nmf318}/J. As the point mutation results in the absence of gene expression, we are referring to the mutant as *Clic5*^{-/-}.

- The poorer functional outcome at later points after early postnatal injection should be discussed: could there be lower transgene expression at these later times?

Response: We thank the reviewer for this important point. Although we did not quantify CLIC5 expression at 12 weeks of age, we observed robust transgene expression, as shown in figure 2A and figure 7A using a specific CLIC5 antibody. In addition, no morphological abnormalities were observed in SEM analysis at 12 weeks post-injection, and vestibular function remained stable at this time point. It is possible that others may influence the ABR thresholds measured, as we describe in the Discussion:

“Limited rescue and durability at high frequencies may result from several factors, including promoter downregulation, an immune response against the viral capsid, the earlier development of the cochlear base compared to other cochlear regions, or the genetic background of the mice, including the *Cdh23* allele in the C57BL/6 strain, which is linked to age-related high-frequency hearing loss (Ivanchenko et al, 2021; Patel et al, 2024; Peters et al, 2023). Despite the deterioration of ABR thresholds over time, transgene expression remained robust at 12 weeks, and no significant morphological changes in hair cells were observed”

We have now clarified this point in the Discussion:

Despite the deterioration of ABR thresholds over time, transgene expression remained robust at 12 weeks, and no significant morphological changes in hair cells were observed.

Specific comments:

- Please make sure to consistently use gene and protein nomenclature (mouse/human) as appropriate for the context

Response: We have reviewed the text and corrected the nomenclature accordingly.

- I suggest to add the age of mice analyzed for Figs. 2 and 3.

Response: We have added the specific ages to the figure legends of Figures 3B–D.

- It did not become entirely clear to me whether the authors differentiate between "restore" and "rescue". Both are used throughout the MS and differentially for figure titles.

Response: We thank the reviewer for pointing this out. We have revised the manuscript to use "rescue" to describe functional improvement and "restore" to refer to the expression or localization of CLIC5.

- This seems to be a fragment: "This outcome has also been observed in the inner ear with AAV vectors driving green fluorescent protein (GFP) expression in cochlear explants and in vivo with cochlear injections (Casey et al, 2020; Maguire & Corey, 2020),..." I also wonder whether the Maguire & Corey review is appropriate as reference here.

Response: We thank the reviewer for pointing it out. Since scAAV has not been widely tested in the inner ear beyond the evaluation of GFP transduction efficiency, we have included the Maguire & Corey reference, which refers to the use of scAAV in cochlear cultures, although the data were not published.

- "DPOAE results also indicated significant differences": specify differences

Response: We have added the following to clarify the differences in DPOAE results:

DPOAE results also indicated significant differences of 20 to 50 dB between the *Clc5*^{-/-} mice and both *Clc5*^{+/+} and *Clc5*^{+/-} groups, except for 6 kHz (Fig 1C; $P \geq 0.13$).

- "Gene therapy restores CLIC5 expression in the auditory system" see general comment

Response: Following the previous comment, we now use "rescue" for functional improvements, and "restore" only for CLIC5 protein re-expression.

- "significantly elongated hair bundles compared to control heterozygous mice, especially in IHCs (Figs 3A and B; $p < 0.0001$)" What measurement does the p value relate to?

Response: The p value relates to the length of the hair bundles when comparing the mutant mice and control heterozygous mice. The statistical test was 2-way ANOVA followed by Holm–Sidak correction for multiple comparison, and the p value ($p < 0.0001$) was consistent across the different regions.

- "Quantification of HC survival indicated that CLIC5 deficiency primarily caused OHC loss" see general comment

Response: We thank the reviewer for this comment. The point mutation results in the absence of gene expression, as reported in Figure 2A. In addition, no expression was detected at earlier developmental stages, confirming it as a deficiency.

- "To evaluate the therapeutic effect of gene replacement therapy on auditory function, ABRs and DPOAEs were recorded at different time points in mice that received injections of ssAAV.Clic5" specify age of injection

Response: We have now added the age at which the injection was done.

- Among the GFP-positive OHCs, the mean fluorescence intensity of scAAV was approximately three times greater than that of ssAAV (28.14 for ssAAV vs. 83.60 for scAAV) (Fig 6D). add arbitrary units!

Response: We have now added a.u. to the description of Fig. 6D to indicate fluorescence intensity measurements.

2nd Jul 2025

Dear Prof. Avraham,

Thank you for the submission of your revised manuscript to EMBO Molecular Medicine. I am pleased to inform you that we will be able to accept your manuscript pending the following final amendments:

- 1) Please implement the referee #2 suggestions.
- 2) Please address all comments suggested by our data editors listed below:
 - o Figure legends:
 1. Please note that the exact p values are not provided in the legends of figures 1A-E, G; 3B-D; 4B, C, D, E, G; 5B, C, F, I, J, L; 6E-G; 7D, E, F, I, J, K, L, N; 8B, C, E, H, I, J, K; EV3 B-D; EV4 A; EV5 A-C.
 2. Please note that information related to n is missing in the legends of figures 1D-G; 3C, D, 4G, 5C, F, G, I, J, K, L; 7E, F, N; 8C, E, F, H, I, J, K; EV2 E-G; EV3 B-D; EV5 A-C.
 3. Please note that the measure of center for the error bars needs to be defined in the legends of figures 1A-G; 3B-D; 4B-G; 5B, C, F, G, I, J, K, L; 7D-F, I-P; 8B, C, E, F, H, I, J, K; EV1 B, C; EV2 A-G; EV3 B-D; EV4 A-D; EV5 A-C.
 - Rename Material and Methods to Methods.
 - Remove "SUPPLEMENTARY MATERIAL", "Expanded view", "Table EV1".
 - Remove Table EV1 legend and place it in the table file. Also, please correct the name of the source file and the title in our submission system to Table EV1.
 - Indicate in legends exact n and exact p values, not a range, along with the statistical test used. To keep the figures "clear" some authors found providing an Appendix table Sx with all exact p-values preferable. You are welcome to do this if you want to.
 - Please place Data availability statement before Acknowledgements and replace the current sentence with "This study includes no data deposited in external repositories."
- 3) Synopsis:
 - Synopsis image: Please resize the image to 550 px-wide x 300-600 pixels high and upload it as a high-resolution jpeg file. Make sure that text is readable and all the items in the image clearly visible.
 - Please check your synopsis text and image before submission with your revised manuscript. Please be aware that in the proof stage minor corrections only are allowed (e.g., typos).
- 4) As part of the EMBO Publications transparent editorial process initiative (see our Editorial at <http://embomolmed.embopress.org/content/2/9/329>), EMBO Molecular Medicine will publish online a Review Process File (RPF) to accompany accepted manuscripts. This file will be published in conjunction with your paper and will include the anonymous referee reports, your point-by-point response and all pertinent correspondence relating to the manuscript. Let us know whether you agree with the publication of the RPF and as here, if you want to remove or not any figures from it prior to publication. Please note that the Authors checklist will be published at the end of the RPF.
- 5) Please provide a point-by-point letter INCLUDING my comments as well as the reviewer's reports and your detailed responses (as Word file).

I look forward to reading a new revised version of your manuscript as soon as possible.

Yours sincerely,

Zeljko Durdevic

Zeljko Durdevic
Senior Editor
EMBO Molecular Medicine

*** Instructions to submit your revised manuscript ***

When submitting your revised manuscript, please

include:

- 1) a .docx formatted version of the manuscript text (including Figure legends and tables)
 - 2) Separate figure files*
 - 3) supplemental information as Expanded View and/or Appendix. Please carefully check the authors guidelines for formatting Expanded view and Appendix figures and tables at <https://www.embopress.org/page/journal/17574684/authorguide#expandedview>
 - 4) a letter INCLUDING the reviewer's reports and your detailed responses to their comments (as Word file).
 - 5) The paper explained: EMBO Molecular Medicine articles are accompanied by a summary of the articles to emphasize the major findings in the paper and their medical implications for the non-specialist reader. Please provide a draft summary of your article highlighting
 - the medical issue you are addressing,
 - the results obtained and
 - their clinical impact.This may be edited to ensure that readers understand the significance and context of the research. Please refer to any of our published articles for an example.
 - 6) Author contributions: the contribution of every author must be detailed in a separate section.
 - 7) EMBO Molecular Medicine now requires a complete author checklist (<https://www.embopress.org/page/journal/17574684/authorguide>) to be submitted with all revised manuscripts. Please use the checklist as guideline for the sort of information we need WITHIN the manuscript. The checklist should only be filled with page numbers where the information can be found. This is particularly important for animal reporting, antibody dilutions (missing) and exact values and n that should be indicated instead of a range.
 - 8) Every published paper now includes a 'Synopsis' to further enhance discoverability. Synopses are displayed on the journal webpage and are freely accessible to all readers. They include a short stand first (maximum of 300 characters, including space) as well as 2-5 one sentence bullet points that summarise the paper. Please write the bullet points to summarise the key NEW findings. They should be designed to be complementary to the abstract - i.e. not repeat the same text. We encourage inclusion of key acronyms and quantitative information (maximum of 30 words / bullet point). Please use the passive voice. Please attach these in a separate file or send them by email, we will incorporate them accordingly.
- You are also welcome to suggest a striking image or visual abstract to illustrate your article. If you do please provide a jpeg file 550 px-wide x 300-600px high.
- 9) A Conflict of Interest statement should be provided in the main text
 - 10) Please note that we now mandate that all corresponding authors list an ORCID digital identifier. This takes <90 seconds to complete. We encourage all authors to supply an ORCID identifier, which will be linked to their name for unambiguous name identification.

Currently, our records indicate that the ORCID for your account is 0000-0002-4913-251X.

Link Not Available

- 11) Include a Reagents and Tools Table as part of the Methods section, which can be downloaded from our author guidelines (<https://www.embopress.org/page/journal/17574684/authorguide#structuredmethods>)

Each figure should be given in a separate file and should have the following resolution:
Graphs 800-1,200 DPI

Photos 400-800 DPI
Colour (only CMYK) 300-400 DPI"

*Additional important information regarding figures and illustrations can be found at <https://bit.ly/EMBOPressFigurePreparationGuideline>. See also figure legend preparation guidelines: <https://www.embopress.org/page/journal/17574684/authorguide#figureformat>

***** Reviewer's comments *****

Referee #1 (Comments on Novelty/Model System for Author):

This is a nicely designed study. The technical quality of the results is high. AAV-mediated gene therapy has been applied to various mouse models of hereditary hearing loss, though this is the first report of its application in a mouse model of DFNB103. The testing of scAAV vs ssAAV is novel and interesting.

Referee #1 (Remarks for Author):

The authors have adequately addressed my comments. I think this study will be a great addition to the growing body of work in the field of inner ear gene therapy.

Referee #2 (Comments on Novelty/Model System for Author):

Study well done

Referee #2 (Remarks for Author):

The revised manuscript by Hahn et al. presents a substantially improved study addressing *Clic5*-related auditory and vestibular dysfunction using AAV-mediated gene delivery.

The authors have addressed nearly all major concerns raised in the initial review. Notably, the addition of dose-response analysis of scAAV, quantification of vestibular hair cell survival, localization of interacting proteins (RDX and TPRN), clarification of the disease timeline via electron microscopy, and GFP transduction data in vestibular organs all strengthen the study's mechanistic and translational relevance.

These additions significantly enhance the robustness of the *Clic5* gene therapy outcomes and their relevance for potential clinical translation.

Minor Comments:

1- Correct typographical inconsistencies:

- Replace "p0 p14" with "P0 and P14".
- Use "myosin VIIa," with a lowercase "m"
- Ensure that *Clic5* is italicized consistently in figure labels

2- For improved clarity and presentation, use consistent color coding for stains across figures-especially for Actin, which currently appears in different colors in various panels (e.g. Fig 2D and E can benefit from using red and green colors, instead of cyan and magenta).

The authors addressed the remaining editorial issues.

9th Jul 2025

Dear Prof. Avraham,

We are pleased to inform you that your manuscript is accepted for publication and is now being sent to our publisher to be included in the next available issue of EMBO Molecular Medicine.

Zeljko Durdevic
Senior Editor
EMBO Molecular Medicine
